# RECONCILING SPATIAL AND TEMPORAL ABSTRACTIONS FOR GOAL REPRESENTATION

**Mehdi Zadem**[1], **Sergio Mover**[1,2], **Sao Mai Nguyen**[3,4]

[1]LIX, École Polytechnique,Institut Polytechnique de Paris, France
[2]CNRS, [3]Flowers Team, U2IS, ENSTA Paris, IP Paris
[4]IMT Atlantique, Lab-STICC, UMR CNRS 6285
{zadem, sergio.mover}@lix.polytechnique.fr
nguyensmai@gmail.com

## ABSTRACT

Goal representation affects the performance of Hierarchical Reinforcement Learning (HRL) algorithms by decomposing the complex learning problem into easier subtasks. Recent studies show that representations that preserve temporally abstract environment dynamics are successful in solving difficult problems and provide theoretical guarantees for optimality. These methods however cannot scale to tasks where environment dynamics increase in complexity i.e. the temporally abstract transition relations depend on larger number of variables. On the other hand, other efforts have tried to use spatial abstraction to mitigate the previous issues. Their limitations include scalability to high dimensional environments and dependency on prior knowledge.

In this paper, we propose a novel three-layer HRL algorithm that introduces, at different levels of the hierarchy, both a spatial and a temporal goal abstraction. We provide a theoretical study of the regret bounds of the learned policies. We evaluate the approach on complex continuous control tasks, demonstrating the effectiveness of spatial and temporal abstractions learned by this approach.[1]

## 1 INTRODUCTION

Goal-conditioned *Hierarchical Reinforcement Learning* (HRL) (Dayan & Hinton, 1992) tackles task complexity by introducing a temporal abstraction over learned behaviours, effectively decomposing a large and difficult task into several simpler subtasks. Recent works (Vezhnevets et al., 2017; Kulkarni et al., 2016; Nachum et al., 2019; Zhang et al., 2023; Li et al., 2021) have shown that learning an abstract goal representations is key to proposing semantically meaningful subgoals and to solving more complex tasks. In particular, representations that capture environment dynamics over an abstract temporal scale have been shown to provide interesting properties with regards to bounding the suboptimality of learned policies under abstract goal spaces (Nachum et al., 2019; Abel et al., 2020), as well as efficiently handling continuous control problems.

However, temporal abstractions that capture aspects of the environment dynamics (Ghosh et al., 2019; Savinov et al., 2019; Eysenbach et al., 2019; Zhang et al., 2023; Li et al., 2021) still cannot scale to environments where the pairwise state reachability relation is complex. For instance, Zhang et al. (2023) compute a k-reachability relation for a subset of the environment's states defined with an oracle (e.g., the oracle selects only the $x, y$ dimensions). While sampling reachable goals is useful to drive efficiency, the learned $k$-adjacency relation is difficult to learn for higher dimensions. This situation typically happens when temporally abstract relations take into account more variables in the state space. The main limitation of these approaches is the lack of a spatial abstraction to generalise such relations over states.

Alternatively, other works (Kulkarni et al., 2016; Illanes et al., 2020; Garnelo & Shanahan, 2019; Zadem et al., 2023) have studied various forms of spatial abstractions for goal spaces. These ab-

---

[1]Find open-source code at https://github.com/cosynus-lix/STAR

stractions effectively group states with similar roles in sets to construct a discrete goal space. The advantage of such representation is a smaller size exploration space that expresses large and long-horizon tasks. In contrast to the algorithms that require varying levels of prior knowledge (Kulkarni et al., 2016; Lyu et al., 2019; Illanes et al., 2020), GARA (Zadem et al., 2023) gradually learns such spatial abstractions by considering reachability relations between sets of states. We refer to this abstraction as *reachability-aware abstraction*. While such representation is efficient for low-dimensional tasks, scalability remains an issue due to the lack of a temporal abstraction mechanism. What is challenging about scaling GARA's approach to more complex environments is exactly what makes the set-based representation effective: the low-level agent must learn how to reach a set of states that, especially in the initial phases of the algorithm when the abstraction is coarser, may be "far" away. We tackle this problem introducing a new agent in the hierarchy that introduces a *temporal abstraction*. Such an agent learns to select intermediate subgoals that: can be reached from a state $s$ executing the low-level agent; and helps constructing a trajectory from $s$ to a goal abstract state.

In this paper, we propose a three-layer HRL algorithm that achieves both a temporal and spatial abstraction for capturing the environment dynamics. We motivate the use of temporal abstraction as the key factor that can scale the abstraction proposed in Zadem et al. (2023), and the reachability-aware spatial abstraction as a way to efficiently represent goals in complex tasks. We complement the approach by adding theoretical guarantees on the bounds of suboptimality of policies learned under this abstraction. Our approach is empirically evaluated on a set of challenging continuous control tasks. Our work presents the following contributions:

(1) A novel Feudal HRL algorithm, STAR, to learn online subgoal representations and policies. STAR consists of 3 agents: the high-level agent selects regions in the abstract reachability-aware goal space, the middle-level agent selects concrete subgoals that help reaching abstract goals, and the low-level agent learns to take actions in the environment (Section 3).

(2) Provide a theoretical motivation for using reachability-aware goal representations, showing a suboptimality bound on the learned policy and that our algorithm progressively improves the reachability-aware abstraction. (Section 4)

(3) Empirically show that STAR successfully combines both temporal and spatial abstraction for more efficient learning, and that the reachability-aware abstraction scales to tasks with more complex dynamics. (Section 5).

## 2 RELATED WORK

Building on ideas for introducing hierarchy in Reinforcement Learning (Sutton et al., 1999; Barto & Mahadevan, 2003; Dayan & Hinton, 1992), recent advances have managed to considerably elevate HRL algorithms to tackle complex continuous control environments. Nachum et al. (2018) introduces a two-level hierarchy that sample goals from a pre-defined oracle on the state space. This approach provides a good basis for HRL algorithms as it is generic and addresses non-stationary learning but may still be suboptimal as the goal sampling is unconstrained in the oracle.

To remedy this, Ghosh et al. (2019); Savinov et al. (2019); Eysenbach et al. (2019); Zhang et al. (2023); Li et al. (2021) learn different goal representations that try to capture the environment dynamics. This idea has been validated under different theoretical formulations (Nachum et al., 2019; Abel et al., 2020; Li et al., 2021). In particular, Li et al. (2021) learns a latent representation based on slow-dynamics in the state space. The idea is that meaningful temporally abstract relations (over $k$ steps) are expressed by state features that slowly change over time. However, these features may not be always sufficient to capture all the critical information about dynamics. Both Savinov et al. (2019) and Zhang et al. (2023) use $k$-step reachability relations as a characterisation for environment dynamics. Their idea is to learn if goals (mappings of states in an embedding / oracle) reach a potential goal in $k$ steps. These relations are later used to drive the sampling of goals that can be reached, resulting in more efficient learning. However, such learned pairwise relations are binary and lack the information of which goals can be reached by applying a specific policy. Additionally, without any spatial abstraction, it can be difficult to learn these relations for a complex transition relation (e.g. a relation that require monitoring more that few variables).

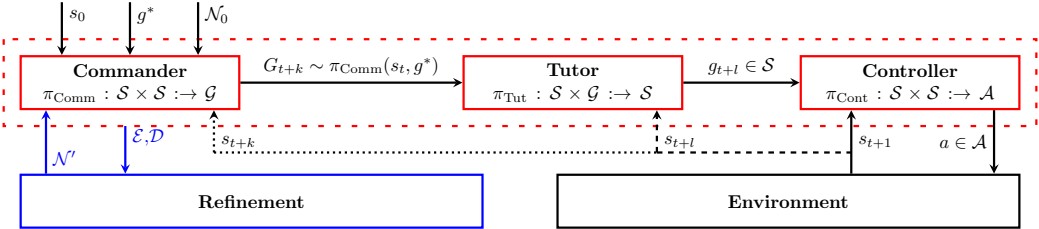

Figure 1: **Architecture of STAR**. The algorithm's inputs are the initial state $s_0$, the task goal $g^*$, and an initial abstraction $\mathcal{N}_0$. STAR runs in a feedback loop a Feudal HRL algorithm (dashed red block) and an abstraction refinement (blue box). The solid red blocks show the HRL agents (*Commander*, *Tutor*, *Controller*). The agents run at different timescales ($k > l > 1$), shown with the solid, dashed, and dotted lines carrying the feedback from the environment to the agents. The Refinement uses as inputs the past episodes ($\mathcal{D}$) and a the list of abstract goals ($\mathcal{E}$) visited during the last episode, and outputs an abstraction.

To introduce spatial abstraction, Zadem et al. (2023) introduce GARA, a spatial abstraction for the goal space that captures richer information from $k$ step reachability relations. This algorithm learns online a discretisation of the state space that serves as an abstract goal space. This abstraction generalizes reachability relations over sets of states, greatly reducing the difficulty of the learning problem. This approach however was only validated on a 4-dimensional environment and suffers from scalability issues as the abstraction is learned concurrently with the hierarchical policies. As GARA starts learning from a coarse abstraction (composed of a small number of large sets), the goal set is often distant from the current state, thus it can be difficult to learn meaningful policies that manage to reach a desired goal set. Under such circumstances, the approach is blocked targeting a hard to reach goal and cannot improve as it lacks any mechanism to propose easier, more granular subgoals. To alleviate this discrepancy, we introduce a new agent in the hierarchy of GARA that applies a *temporal abstraction* (Sutton et al., 1999). Our intuition is to synthesise between the discrete goal chosen from the top-down process and the temporal transitions allowed by the current low-level policy from the bottom-up process, through a mid-level agent that acts as an intelligent tutoring system and learns to select intermediate goals $g \in \mathcal{S}$ that: (a) can be reached from a state $s$ executing the current low-level agent; and (b) helps constructing a trajectory from $s$ to a goal abstract state $G \in \mathcal{G}$.

## 3 SPATIO-TEMPORAL ABSTRACTION VIA REACHABILITY

We consider a goal-conditioned Markov Decision Process $(\mathcal{S}, \mathcal{A}, P, r_{ext})$ as environment, where $\mathcal{S} \subseteq \mathbb{R}^n$ is a continuous state space, $\mathcal{A}$ is an action space, $P(s_{t+1}|s_t, a_t)$ is a probabilistic transition function, and $r_{ext} : \mathcal{S} \times \mathcal{S} \to \mathbb{R}$ is a parameterised reward function, defined as the negative distance to the task goal $g^* \in \mathcal{S}$, i.e $r_{ext}(s, g^*) = -\|g^* - s\|_2$. The *multi-task reinforcement learning problem* consists in learning a goal conditioned policy $\pi$ to sample at each time step $t$ an action $a \sim \pi(s_t \mid g^*)$, so as to maximizes the expected cumulative reward. The spatial goal abstraction is modeled by a *set-based abstraction* defined by a function $\mathcal{N} : \mathcal{S} \to 2^{\mathcal{S}}$ that maps concrete states to sets of states (i.e., $\forall s \in \mathcal{S}, \mathcal{N}(s) \subseteq \mathcal{S}$). We write $\mathcal{G}_{\mathcal{N}}$ to refer to the range of the abstraction $\mathcal{N}$, which is intuitively the abstract goal space. We further drop the subscript (i.e., write $\mathcal{G}$) when $\mathcal{N}$ is clear from the context and denote elements of $\mathcal{G}$ with the upper case letter $G$.

We propose a HRL algorithm, Spatio-Temporal Abstraction via Reachability (STAR), that learns, at the same time, a spatial goal abstraction $\mathcal{N}$ and policies at multiple time scales. The STAR algorithm, shown in Figure 1, has two main components: a 3-levels Feudal HRL algorithm (enclosed in the red dashed lines); and an abstraction refinement component (shown in the blue solid lines). STAR runs the Feudal HRL algorithm and the abstraction refinement in a feedback loop, refining the abstraction $\mathcal{N}$ at the end of every learning episode. Observe that the high-level agent (called *Commander*) samples a goal $G$ from the abstract goal space $\mathcal{G}$, and that such a goal can be difficult to reach from the current state. The first intuition of the algorithm is that, to reach a possibly very far goal $G$ set by the high-level agent (*Controller*), the middle-level agent (*Tutor*) achieves a temporal abstraction of the *Controller* actions by sampling an intermediate subgoal $g \in \mathcal{S}$ of a difficulty

level adapted to the current non-optimal policy. The second intuition is that the algorithm learns the abstraction iteratively. Every refinement obtains a finer abstraction $\mathcal{N}'$ from $\mathcal{N}$. Intuitively, $\mathcal{N}'$ will split at least a goal $G_1 \in \mathcal{G}_\mathcal{N}$ in two goals $G'_1, G''_1 \in \mathcal{G}_{\mathcal{N}'}$ if there are different states in $G_1$ (i.e., $s_a, s_b \in G_1$) that cannot reach the same target $G_2 \in \mathcal{G}$ when applying the same low-level policy. While we will define such *reachability property* precisely later, intuitively the refinement separates goal states that "behave differently" under the same low level policy (i.e., $\mathcal{N}'$ would represent more faithfully the environment dynamic).

We first explain how STAR uses the different hierarchy levels to address the challenges of learning policies when the abstraction is coarse (Section 3.1), and then formalize a refinement that obtains a *reachability aware* abstraction (Section 3.2).

### 3.1    A 3-LEVEL HIERARCHICAL ALGORITHM FOR TEMPORAL ABSTRACTION

The feudal architecture of STAR is composed of a hierarchy with three agents:

1. *Commander*: the highest-level agent learns the policy $\pi_{\text{Comm}} : \mathcal{S} \times \mathcal{S} \to \mathcal{G}$ that is a goal-conditioned on $g^*$ and samples an abstract **goal** $G \in \mathcal{G}$ that should help to reach the task goal $g^*$ from the current agent's state ($G_{t+k} \sim \pi_{\text{Comm}}(s_t, g^*)$).
2. *Tutor*: the mid-level agent is conditioned by the *Commander* goal $G$. It learns the policy $\pi_{\text{Tut}} : \mathcal{S} \times \mathcal{G} \to \mathcal{S}$ and picks **subgoals** in the state space ($g_{t+l} \sim \pi_{\text{Tut}}(s_t, G_{t+k})$).
3. *Controller*: the low-level policy $\pi_{\text{Cont}} : \mathcal{S} \times \mathcal{S} \to \mathcal{A}$ is goal-conditioned by the *Tutor*'s subgoal $g$ and samples actions to reach given goal ($a \sim \pi_{\text{Cont}}(s_t, g_{t+l})$). [2]

The agents work at different time scales: the *Commander* selects a goal $G_{t+k}$ every $k$ steps, the *Tutor* selects a goal $g_{t+l}$ every $l$ steps (with $k$ multiple of $l$), and the *Controller* selects an action to execute at every step. Intuitively, the *Commander*'s role in this architecture is to select an abstract goal $G_{t+k}$ (i.e. a set of states that are similar and of interest to solve the task) from the current state $s_t$. However, the initial abstraction $\mathcal{G}$ of the state space is very coarse (i.e., the abstract regions are large sets and still do not represent the agent's dynamic). This means that learning a flat policy that reaches $G_{t+k}$ from $s_t$ is very challenging. The *Tutor* samples subgoals $g_{t+l}$ from $\mathcal{S}$ that are intermediate, easier targets to learn to reach for the *Controller*, instead of a possibly far away state that might prove too difficult to reach. Intuitively, the *Tutor* implements a *temporal abstraction mechanism*. The structure of STAR guides the agent through the large state-goal abstraction, while it also allows the agent to learn manageable low-level policies.

We set the reward at each level of the hierarchy following the above intuitions. The *Commander* receives the external environment reward after $k$ steps for learning to reach the task goal $g^*$: $r_{\text{Comm}}(s_t) := \max_{x \in \mathcal{N}(s_t)} r_{ext}(x, g^*)$ . This reward is computed as the extrinsic reward of the closest point in $\mathcal{N}(s_t)$ to $g^*$. The *Tutor* helps the *Controller* to reach the abstract goal, and is thus rewarded by a distance measure between $s_t$ and the $G_{t+k}$; $r_{\text{Tut}}(s_t, G_{t+k}) := -\|s_t - \text{Center}(G_{t+k})\|_2$, where $\text{Center}(G_{t+k})$ is the center of the goal $G_{t+k}$. Finally, the *Controller* is rewarded with respect to the Euclidean distance between subgoals sampled by the *Tutor* and the reached state: $r_{\text{Cont}}(s_t, g_{t+l}) := -\|g_{t+l} - s_t\|_2$. This intrinsic reward allows the *Controller* to learn how to reach intermediate subgoal states $g_{t+l}$.

### 3.2    REFINING $\mathcal{N}$ VIA REACHABILITY ANALYSIS

While we follow the high level description of the refinement procedure of the GARA (Zadem et al., 2023) algorithm, we adapt it to our notation and to the new theoretical results on the refinement we present later (which holds for both GARA and STAR). Furthermore, in the rest of this Section and in Section 4, we will assume a 2-level hierarchy, where $\pi_{\text{Comm}}$ is the high-level policy (e.g., $\pi_{\text{high}}$), and $\pi_{\text{low}}$ is the hierarchical policy obtained composing $\pi_{\text{Tut}}$ and $\pi_{\text{Cont}}$.

We first define the *$k$-step reachability relation* for a goal-conditioned policy $\pi_{\mathcal{N}_{\text{Low}}}$:

$$R^k_{\pi_{\mathcal{N}_{\text{Low}}}}(G_i, G_j) := \left\{ s' \in \mathcal{S} \mid s \in G_i, s \xrightarrow[\pi_{\mathcal{N}_{\text{Low}}}(., G_j)]{k} s' \right\},$$

---

[2] In the following, we use the upper-case $G$ letter for goals in $\mathcal{G}$ and the lower-case $g$ for subgoals in $\mathcal{S}$.

where $s \xrightarrow[\pi_{\mathcal{N}_{Low}}(.,G_j)]{k} s'$ means that $s$ can reach $s'$ by executing the policy $\pi_{\mathcal{N}_{Low}(.,G_j)}$ (targeting $G_j$) in $k$ steps. In other words, $R^k_{\pi_{\mathcal{N}_{Low}}}(G_i, G_j)$ is the set of states reached when starting in any state in $G_i$ and applying the policy $\pi_{\mathcal{N}_{Low}}(., G_j)$ for $k$ steps.

The algorithm uses the notion of reachability property among a pair of abstract goals:

**Definition 1 (Pairwise Reachability Property)** *Let $\mathcal{N} : \mathcal{S} \to 2^{\mathcal{S}}$ be a set-based abstraction and $G_i, G_j \in \mathcal{G}_{\mathcal{N}}$. $\mathcal{N}$ satisfies the pairwise reachability property for $(G_i, G_j)$ if $R^k_{\pi_{\mathcal{N}_{Low}}^*}(G_i, G_j) \subseteq G_j$.*

The algorithm decides to refine the abstract representation after an episode of the HRL algorithm. Let $\mathcal{E} := \{G_0, \ldots, G_n\}$ be the list of goals visited in the last episode. The refinement algorithm analyzes all the pairs $(G_i, G_{i+1}) \in \mathcal{E}$, for $0 \leq i < n$, and refines $\mathcal{N}$ in a new abstraction $\mathcal{N}'$ by "splitting" $G_i$ if it does not satisfy the pairwise reachability property. Each refinement obtains a new, finer abstraction $\mathcal{N}'$ where the reachability property is respected in one more goal. We formalize when an abstraction $\mathcal{N}'$ refines an abstraction $\mathcal{N}$ with respect to the reachability property as follows:

**Definition 2 (Pairwise Reachability-Aware Refinement)** *Let $\mathcal{N} : \mathcal{S} \to 2^{\mathcal{S}}$ and $\mathcal{N}' : \mathcal{S} \to 2^{\mathcal{S}}$ be two set-based abstractions such that there exists $G_i \in \mathcal{G}_{\mathcal{N}}$, $\mathcal{G}_{\mathcal{N}'} = (\mathcal{G}_{\mathcal{N}} \setminus \{G_i\}) \cup \{G'_1, G'_2\}$, $G'_1 \cup G'_2 = G_i$, and $G'_1 \cap G'_2 = \emptyset$. $\mathcal{N}'$ refines $\mathcal{N}$ (written as $\mathcal{N}' \prec \mathcal{N}$) if, for some $G_j \in \mathcal{G}_{\mathcal{N}}$, $\mathcal{N}'$ satisfies the pairwise reachability property for $(G'_1, G_j)$, while $\mathcal{N}$ does not satisfy the pairwise reachability property for $(G_i, G_j)$.*

### 3.2.1 REFINEMENT COMPUTATION

We implement the refinement similarly to GARA. We represent an abstraction $\mathcal{N}$ directly with the set of abstract states $\mathcal{G}_{\mathcal{N}} := \{G_1, \ldots, G_n\}$, a partition of the state space $\mathcal{S}$ (i.e., all the sets in $\mathcal{G}_{\mathcal{N}}$ are disjoint and their union is $\mathcal{S}$). We represent each $G \in \mathcal{G}_{\mathcal{N}}$ as a multi-dimensional interval (i.e., a hyper-rectangle). We (i) train a neural network $\mathcal{F}_k$ predicting the k-step reachability from each partition in $\mathcal{G}_{\mathcal{N}}$; and (ii) for each $G_i, G_{i+1} \in \mathcal{E}$, we check pairwise reachability from $\mathcal{F}_K$; and (iii) if pairwise reachability does not hold, we compute a refinement $\mathcal{N}'$ of $\mathcal{N}$.

We approximate the reachability relation with a *forward model*, a fully connected feed forward neural network $\mathcal{F}_k : \mathcal{S} \times \mathcal{G}_{\mathcal{N}} \to \mathcal{S}$. $\mathcal{F}_k(s_t, G_j)$, is trained from the replay buffer and predicts the state $s_{t+k}$ the agent would reach in $k$ steps starting from a state $s_t$ when applying the low level policy $\pi_{\mathcal{N}_{Low}}(s, G_j)$ conditioned on the goal $G_j$. We avoid the *non-stationarity* issue of the lower-level policy $\pi_{\mathcal{N}_{Low}}$ by computing the refinement only when $\pi_{\mathcal{N}_{Low}}$ is *stable*. See Appendix C for more details. We use the forward model $\mathcal{F}_k$ to evaluate the policy's stability in the regions visited in $\mathcal{E}$ by computing the Mean Squared Error of the forward model predictions on a batch of data from the replay buffer, and consider the model stable only if the error remains smaller than a parameter $\sigma$ over a window of time.

Checking the reachability property for $(G_i, G_j)$ amounts to computing the output of $\mathcal{F}_k(s, G_j)$, for all states $s \in G_i$, i.e., $R^k_{\pi_{\mathcal{N}_{Low}}}(G_i, G_j) := \{\mathcal{F}_k(s, G_j) \mid s \in G_i\}$, and checking if $R^k_{\pi_{\mathcal{N}_{Low}}}(G_i, G_j) \subseteq G_j$. Technically, we compute an *over-approximation* $\tilde{R}^k_{\pi_{\mathcal{N}_{Low}}}(G_i, G_j) \supseteq R^k_{\pi_{\mathcal{N}_{Low}}}(G_i, G_j)$ with the Ai2 (Gehr et al., 2018) tool. Ai2 uses abstract interpretation (Cousot & Cousot, 1992; Rival & Yi, 2020) to compute such an over-approximation: the tool starts with a set-based representation (e.g., intervals, zonotopes, ...) of the neural network input, the set $G_i$, and then computes the set of outputs layer-by-layer. In $F_k$, each layer applies first a linear transformation (i.e. for the weights and biases) and then a piece-wise ReLU activation function. For each layer, Ai2 computes first an over-approximation of the linear transformation (i.e., it applies the linear transformation to the input set), and then uses this result to compute an over-approximation of the ReLU activation function, producing the new input set to use in the next layer. Such computations work on abstract domains: for example, one can over-approximate a linear transformation applied to an interval by computing the transformation, and then a convex-hull to obtain the output in the form of an interval. The over-approximations for ReLU activation functions are described in detail in Gehr et al. (2018).

We split the set $G_i$ when it does not satisfy the reachability property w.r.t. $G_{i+1}$. Since $G_i$ is an interval, we implement the algorithm from (Zadem et al., 2023; Wang et al., 2018) that: (i) stops if

$G_i$ satisfies the reachability property or none of the states in $G_i$ reach $G_j$; and otherwise (ii) splits $G_i$ in two intervals, $G'_i$ and $G''_i$, calling the algorithm recursively on both intervals. This results in two sets of intervals, subsets of $G_i$, that either satisfy the reachability property or not. Such intervals are the new intervals replacing $G_i$ in the new abstraction (see Appendix F).

## 4 THEORETICAL PROPERTIES OF THE REFINEMENT

In this section, we motivate the adoption of the goal-space abstraction and the reachability-aware refinement showing that: (i) there exists a bound on the sub-optimality of policies trained with a reachability-aware abstraction; and (ii) the reachability-aware refinement gradually finds a reachability-aware abstraction. Our results apply to both STAR and GARA (Zadem et al., 2023), and all the proofs are available in the Appendix A.

The theoretical results hold under the assumptions that the environment $M$ is *deterministic* and the reward signal $r_{ext}$ is bounded in the environment. Consequently, we assume that the distance separating a state $s \in \mathcal{S}$ from all the states $s' \in \mathcal{S}$ that $s$ can reach in one step is bounded. Thus, there is an upper bound $R_{\max} := \max_{s,s' \in \mathcal{S}, \sum_{a \in \mathcal{A}} P(s'|s,a) \geq 0} \|s - s'\|_2$ on the reward signal.

Let $\pi^*$ be the optimal hierarchical policy composed by a high-level policy $g \sim \pi^*_{\text{high}}(s, g^*)$ that samples $g \in \mathcal{S}$, and a low-level policy $a \sim \pi^*_{\text{low}}(s, g)$ that samples actions $a \in \mathcal{A}$. Since the environment is deterministic, there exists an *optimal high-level trajectory* containing the *goals* sampled with $\pi^*_{\text{high}}$ and an *optimal low-level trajectory* containing all the visited states:

$$\mathcal{T}^*_{\text{high}} := \{g_0, g_1, \ldots, g_m\}, \quad \mathcal{T}^*_{\text{Low}} := \{s_0, s_1, \ldots, s_{m \cdot k}\}, \text{ with } s_{i \cdot k} = g_i, \quad \text{for } 0 \leq i \leq m.$$

Let $\mathcal{N} : \mathcal{S} \to 2^{\mathcal{S}}$ be a set-based abstraction. We write $\pi^*_{\mathcal{N}}$ for the *optimal* hierarchical policy obtained with the abstraction $\mathcal{N}$. We write $\mathcal{T}^*_{\mathcal{N}_{\text{High}}}$ and $\mathcal{T}^*_{\mathcal{N}_{\text{Low}}}$ for the optimal high- and low-level trajectories respectively. Below, we provide an upper bound on the difference between the optimal hierarchical policy $\pi^*$ and the optimal hierarchical policy $\pi^*_{\mathcal{N}}$ when $\mathcal{N}$ is a *reachability-aware*.

**Definition 3 (Reachability-Aware Abstraction)** *Let $\mathcal{N} : \mathcal{S} \to 2^{\mathcal{S}}$ be a set-based abstraction, $\pi^*_{\mathcal{N}}$ the corresponding optimal hierarchical policy, and $\mathcal{T}^*_{high}$ the optimal high-level trajectory from $\pi^*_{high}$. $\mathcal{N}$ is a reachability-aware abstraction with respect to $\mathcal{T}^*_{high}$ if:*

1. *States are contained in their abstraction: $\forall s \in \mathcal{S}, s \in \mathcal{N}(s)$.*

2. *The abstractions of the goals in the optimal trajectory are disjoint:*
$$\forall g_i, g_j \in \mathcal{T}^*_{high}, \ (g_i \neq g_j \to \mathcal{N}(g_i) \cap \mathcal{N}(g_j) = \emptyset).$$

3. *The abstractions of each consecutive goals in the optimal trajectory satisfy the pairwise reachability property:*
$$\forall g_i, g_{i+1} \in \mathcal{T}^*_{high}, R^k_{\pi^*_{\mathcal{N}_{Low}}}(\mathcal{N}(g_i), \mathcal{N}(g_{i+1})) \subseteq \mathcal{N}(g_{i+1}).$$

4. *The reward in the final abstract goal $\mathcal{N}(g_m)$ is bounded:*
$$\exists \epsilon > 0, \forall s \in \mathcal{N}(g_m).|r_{ext}(g_m) - r_{ext}(s)| \leq \epsilon.$$

**Theorem 1 (Sub-optimal Learning)** *Let $M$ be a deterministic environment with task goal $g^* \in \mathcal{S}$ and $r_{ext}(s) = -\|g^* - s\|_2$. Let $\mathcal{N} : \mathcal{S} \to 2^{\mathcal{S}}$ be a reachability-aware abstraction with respect to $\mathcal{T}^*_{high}$. Then, for $s_0 \in \mathcal{T}^*_{Low}$ and $s'_0 \in \mathcal{T}^*_{\mathcal{N}_{Low}}$ we have that:*

$$|V_{\pi^*}(s_0) - V_{\pi^*_{\mathcal{N}}}(s'_0)| \leq \left( \sum_{i=0}^{\frac{mk}{2}} \gamma^i i + \sum_{i=\frac{mk}{2}}^{mk} \gamma^i (mk - i) \right) \cdot 2R_{max} + \frac{1 - \gamma^{mk+1}}{1 - \gamma} \epsilon, \quad (1)$$

*where $V_\pi(s)$ is the value function for a policy $\pi$ (Sutton & Barto, 1998).*
*Moreover, if there exists a $B \geq \epsilon > 0$ such that for all $1 \leq i \leq m$, $\max_{x,y \in \mathcal{N}(g_i)} \|x - y\| \leq B$, then $\forall s_i \in \mathcal{T}^*_{Low}$ and $\forall s'_i \in \mathcal{T}^*_{\mathcal{N}_{Low}}$ we have that:*

$$|V_{\pi^*_{Low}}(s_0) - V_{\pi^*_{\mathcal{N}_{Low}}}(s'_0)| \leq \frac{1 - \gamma^{mk+1}}{1 - \gamma}(kR_{max} + B). \quad (2)$$

Equation (1) in the above theorem provides a bound on the sub-optimality of the hierarchical policy when trained under a set-based reachability-aware abstraction $\mathcal{N}$. Intuitively, the worst trajectory $\mathcal{T}^*_{\mathcal{N}_{\text{Low}}}$, starting from $s'_0 \in \mathcal{N}(s_0)$, can progressively deviate from $\mathcal{T}^*_{\text{Low}}$ as $i$ increases. When $i \geq \frac{mk}{2}$, the trajectories progressively converge around $\mathcal{N}(g_m)$. Equation (2) defines a tighter upper bound when there is a bound $B$ on the maximum distance between two states in each abstract goal in $\mathcal{T}^*_{\mathcal{N}_{\text{High}}}$. In practice, the existence of the bound $B$ is valid when the state space is bounded. In this case, the deviation of the two trajectories is independent from $i$ and is stable across time.

**Lemma 1** *Let $\mathcal{N}$ and $\mathcal{N}'$ be two set-based abstractions such that $\mathcal{N}' \prec \mathcal{N}$ and $\mathcal{N}$ satisfies the Conditions (1), and (4) of a reachability-aware abstraction (Definition 3). Also, let $G_i \in \mathcal{T}^*_{\mathcal{N}_{\text{High}}}$ (note that $G_i \in \mathcal{G}_{\mathcal{N}}$), and $G_i$ be the goal refined in $\mathcal{N}'$. Then, the abstraction $\mathcal{N}'$ satisfies the following:*

1. $\exists\, g_i \in \mathcal{T}^*_{high}$ *such that $\mathcal{N}$ does not satisfy the reachability property for $(\mathcal{N}(g_i), \mathcal{N}(g_{i+1}))$, while $\mathcal{N}'$ does for $(\mathcal{N}'(g_i), \mathcal{N}(g_{i+1}))$.*

2. *If there exists $g_j \in \mathcal{T}^*_{high}$ such that $g_j \in \mathcal{N}(g_i)$, then $g_j \notin \mathcal{N}'(g_i)$.*

**Theorem 2** *Given an initial set-based abstraction $\mathcal{N}$ and assuming $\mathcal{N}(g_m)$ satisfies the Conditions (1), and (4) of Definition 3, we compute a reachability-aware abstraction after applying a finite number of reachability-aware refinements.*

Theorem 2 follows from Lemma 1 and shows that applying the reachability-aware refinement for a finite number of times we compute a reachability-aware abstraction. In practice, the assumption that $\mathcal{N}(g_m)$ verifies criteria 1 and 4 of Def.3 is reasonable since the goal $g^*$ is known in the environment. That is to say, $\mathcal{N}(g_m)$ could correspond to a region whose center is $g^*$ and radius is $R_{max}$.

## 5 EXPERIMENTAL EVALUATION

In this section we answer the following research questions: 1) Do the spatial and temporal abstraction of STAR allow for more data-efficient learning? 2) Does the reachability-aware abstraction scale to more complex environments compared to a more concrete reachability relation? 3) How does the learned abstraction decompose the environment to allow for learning successful policies?

### 5.1 ENVIRONMENT SETUP

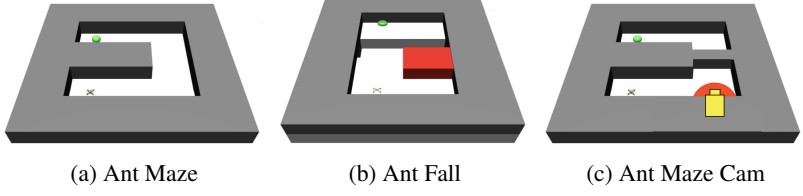

(a) Ant Maze      (b) Ant Fall      (c) Ant Maze Cam

Figure 2: Ant environments

We evaluate our approach on a set of challenging tasks in the Ant environments (Fig.2) adapted from Duan et al. (2016) and popularised by Nachum et al. (2018). The Ant is a simulated quadrupedal robot whose body is composed of a torso and four legs attached by joints. Furthermore, each leg is split into two parts by a joint in the middle to allow bending. The observable space of the Ant is composed of the positions, orientations and velocities of the torso as well as the angles and angular velocities in each joint. Overall the state space is comprised of 30 dimensions. The actions correspond to applying forces on actuators in the joint. This actions space is continuous and 8-dimensional. We propose the following tasks:

1. **Ant Maze:** in this task, the ant must navigate a '⊃'-shaped maze to reach the exit positioned at the top left.

2. **Ant Fall:** the environment is composed of two raised platforms seperated by a chasm. The ant starts on one of the platforms and must safely cross to the exit without falling. A movable block can be push into the chasm to serve as a bridge. Besides the precise maneuvers required by the ant, falling into the chasm is a very likely yet irreversible mistake.

3. **Ant Maze Cam:** this is a more challenging version of Ant Maze. The upper half of the maze is fully blocked by an additional obstacle that can only be opened when the ant looks at the camera (in yellow in Fig. 2c) when on the red spot. The exit remains unchanged.

We note that these tasks are hierarchical in nature as reaching the goal requires correctly controlling the ant to be able to move (low-level) and then navigating to the exit (high-level). Moreover, in both Ant Maze Cam and Ant Fall, no reward is attributed to the intermediate behaviour that unlocks the path to the exit (looking at the camera, pushing the block). Under such circumstances, the underlying dynamics are more complex making the reachability relations more difficult to extract.

## 5.2 COMPARATIVE ANALYSIS

We compare STAR with the following algorithms:

1. **GARA (Zadem et al., 2023)**: this algorithm learns a spatial abstraction via reachability analysis using a two-level hierarchical policy.

2. **HIRO (Nachum et al., 2018)**: this algorithm relies on a Manager to sample goals directly from the state space $\mathcal{S}$ and learns how to achieve them using the controller.

3. **HRAC (Zhang et al., 2023)**: adopting the same architecture as HIRO, this approach tries to approximate a reachability relation between goals in an abstract space and use it to sample rewarding reachable goals. The reachability relation is derived from measuring the shortest transition distance between states in the environment.

4. **LESSON (Li et al., 2021)**: a HRL algorithm that learns a latent goal representations based on slow dynamics in the environment. The latent space learns from features that are slow to change over $k$ steps, in order to capture a temporal abstraction.

In line with HIRO and HRAC, STAR relies on an oracle $\psi(s)$ that transforms the observations of the high-level agents (*Tutor* and *Commander*). In practice $\psi()$ corresponds to a feature selection applied to states. In contrast, LESSON learns a latent goal space without an oracle. In Ant Maze $\psi(s) = (x, y)$, in Ant Fall $\psi(s) = (x, y, z)$, and in Ant Maze Cam, $\psi(s) = (x, y, \theta_x, \theta_y, \theta_z)$.

Fig.3 shows that STAR outperforms all of the state-of-art approaches by reaching a higher success rate with less timesteps. In particular GARA, operating only under a spatial abstraction mechanism is unable to solve Ant Maze, the easiest task in this analysis. HIRO on the other hand learns less efficient policies due to it lacking a spatial abstraction component. These results show that STAR, which combines temporal and spatial abstractions, is a more efficient approach.

To discuss the second research question, we first observe that, while the high-level dynamics of Ant Maze can be captured by the $x, y$ dimensions, the dynamics of Ant Fall require all the $x, y, z$ dimensions ($z$ expresses if the ant is safely crossing above the pit or if it has fallen), and Ant Maze Cam requires $x, y, \theta_x, \theta_y$, and $\theta_z$ (the orientation angles are necessary to unlock the access to the upper part of the maze). Fig.3 shows that HRAC is unable to capture meaningful relations between subgoals and fails at solving either Ant Fall or Ant Maze Cam due to the increased complexity in capturing the high-level task dynamic. Similarly, LESSON is unable to learn a good subgoal representation using slow dynamics in Ant Maze Cam. In fact, $\theta_x, \theta_y$, and $\theta_z$ are features that do not respect the LESSON's slowness objective (i.e., they can vary rapidly across $k$ steps). As a results, the goal abstraction in LESSON may overlook them, losing critical information in the process. Instead, STAR is capable of abstracting these dimensions and converging to a successful policy. We remark that STAR, similarly to HRAC, can be seen as extensions of HIRO with the addition of a reachability-based component that improves goal representation. However, the results shown in Fig. 3 highlight how the addition of the reachability information in HRAC is even detrimental for the performance when the number of features in the oracle increases (e.g., on Ant Fall and Ant Maze Cam). Instead, the STAR's reachability-aware spatial abstraction and intermediate temporal abstraction allow the algorithm to scale to more complex tasks.

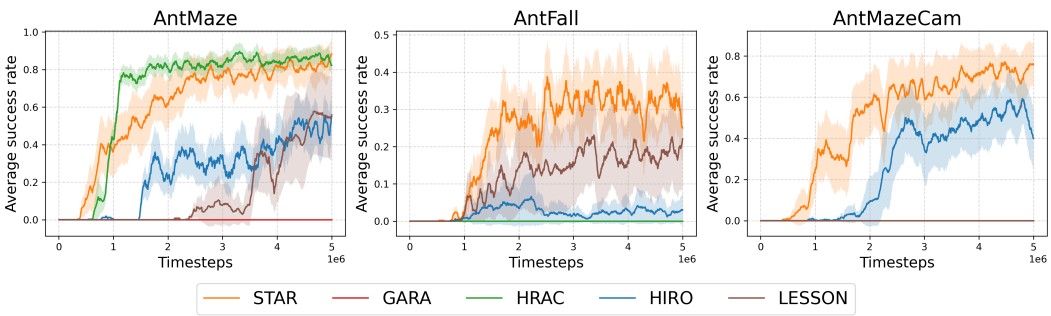

Figure 3: Comparative evaluation averaged over 10 runs for STAR, GARA, HRAC, HIRO and LESSON.

### 5.3 REPRESENTATION ANALYSIS

We answer the third research question examining the progress of the STAR's *Commander* agent at different timesteps during learning when solving the Ant Maze. From Fig.4 we can see that, progressively, the ant explores trajectories leading to the goal of the task. Additionally, the frequency of visiting goals in the difficult areas of the maze (e.g., the tight corners) is higher, and these goals are eventually refined later in the training, jibing with the configuration of the obstabcles. Note that the *Commander*'s trajectory at 3M timesteps sticks close to the obstacles and pass through the maze's opening, resembling an optimal trajectory. This study provides some insight on how STAR gradually refines the goal abstraction to identify successful trajectories in the environment. In particular, STAR learns a more precise abstraction in bottleneck areas where only a few subset of states manage to reach the next goal. We provide the representation analysis on Ant Fall in Appendix B.

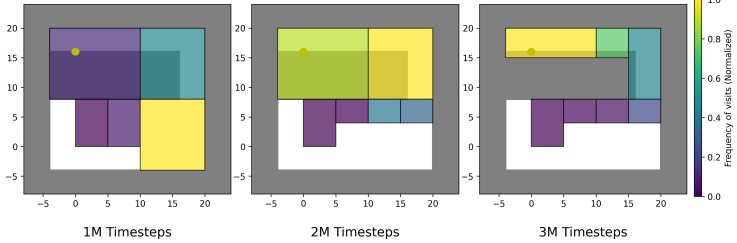

Figure 4: Frequency of goals visited by the *Commander* when evaluating a policy learned after 1M, 2M, and 3M timesteps (averaged over 5 different evaluations with 500 maximum timesteps). The subdivision of the mazes represent (abstract) goals. The color gradient represents the frequency of visits of each goal. Grey areas correspond to the obstacles of the environment in Ant Maze.

### 6 CONCLUSION

In this paper, we propose a novel goal-conditioned HRL algorithm, STAR, that combines spatial and temporal abstractions. The spatial representation groups states with similar environment dynamics, and the temporal abstraction compensates for the non-optimality of the low-level policy, allowing online learning of both the policies and the representations. STAR's high-level agent learns a coarse, although precise, spatial approximation of the environment dynamics that, differently from other algorithms using reachability for goal representation, scales to more complex continuous control environments. STAR outperforms the existing algorithms that use either one of the abstractions and that struggle to learn meaningful representations. Here, we further provide a theoretical justification for reachability-aware abstractions, showing that we can use such representations to learn sub-optimal policies and learn them with a sequence of refinements. In the future, we plan to extend our approach to stochastic environments, which would require a different reachability property, and non-Markovian environments, which would require adding a history component to the spatial abstraction.

ACKNOWLEDGMENTS

We thank the anonymous reviewers for their detailed feedback and discussion on our work.

This research work was partially supported by the Hi! PARIS Center, the Labex DigiCosme (project ANR11LABEX0045DIGICOSME) operated by ANR as part of the program "Investissement d'Avenir" Idex ParisSaclay (ANR11IDEX000302), the SAIF project (funded by the "France 2030" government investment plan managed by the French National Research Agency under the reference ANR-23-PEIA-0006), and the AID/CIEDS project FARO.

We additionally extend our thanks to the members of the VISTA team in LIX, Ecole Polytechnique for their generous support and provision of computational resources, which were instrumental in conducting the experiments for this research.

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

# A PROOFS OF THEOREMS

## A.1 PROOF OF THEOREM 1:

In this part of the proof, we will study the optimality of the hierarchical policies across the last $k$ steps of the trajectory. First, we consider the optimal trajectory for a flat (non-hierarchical policy) in the environment described by $M$:

$$\mathcal{T}^* := \{s_0, s_1, \ldots, s_M\}, \text{ where } s_M = g^*$$

The optimal hierarchical policy $\pi^*$ composed by $\pi^*_{\text{High}}$ and $\pi^*_{\text{Low}}$ samples a goal $g \in \mathcal{S}$ every $k$ steps (where $k$ is a parameter). The following optimal trajectories are thus:

$$\mathcal{T}^*_{\text{High}} := \{g_0, g_1, \ldots, g_m\}, \text{ where } g_m = g^*$$

$$\mathcal{T}^*_{\text{Low}} := \{s_0, s_1, \ldots, s_{mk}\}, \text{ where } s_{n \cdot k} = g_n, n \in [\![0, \ldots, m]\!] \text{ and } s_{mk} = g^* {}^{3}$$

The optimal hierarchical policy $\pi^*_{\mathcal{N}}$ under the abstraction $\mathcal{N}$ (defined by Def.3) yields the optimal trajectories:

$$\mathcal{T}^*_{\mathcal{N}_{\text{High}}} := \{\mathcal{N}(g_0), \mathcal{N}(g_1), \ldots, \mathcal{N}(g_m)\}$$

$$\mathcal{T}^*_{\mathcal{N}_{\text{Low}}} := \{s'_0, s'_1, \ldots, s'_{mk}\}, \text{ where } s'_{n \cdot k} \in \mathcal{N}(g_n), n \in [\![0, \ldots, m]\!] \text{ and } s'_{mk} \in \mathcal{N}(g_m)$$

We make the distinction between $s$ and $s'$ since $\pi^*_{\text{Low}}$ and $\pi^*_{\mathcal{N}_{\text{Low}}}$ do not necessarily follow the same trajectory. Whereas $\pi^*_{\text{Low}}$ is conditioned by goals that act as points in the state space, $\pi^*_{\mathcal{N}_{\text{Low}}}$ is conditioned by a whole set of points. Moving forward, we specify the following rewards functions for the hierarchical policies learning without abstraction and with the abstraction $\mathcal{N}$:

$$\text{For } \pi^*, r_{\text{High}}(s, g^*) = r_{\text{ext}}(s) = -\|g^* - s\|_2 \text{ and } r_{\text{Low}}(s, g) = -\|g - s\|_2$$

$$\text{For } \pi^*_{\mathcal{N}}, r_{\mathcal{N}_{\text{High}}}(s, g^*) = \max_{x \in \mathcal{N}(s)} r_{\text{ext}}(x, g^*) = \max_{x \in \mathcal{N}(s)} -\|g^* - x\|_2 \text{ and } r_{\mathcal{N}_{\text{Low}}}(s, \mathcal{N}(g)) = \mathbb{1}_{x \in \mathcal{N}(g)}(s).$$

Intuitively, $r_{\mathcal{N}_{\text{High}}}(s)$ is computed over the closest point to $g^*$ in $\mathcal{N}(s)$. $r_{\mathcal{N}_{\text{Low}}}(s, \mathcal{N}(g))$ is a binary reward that expresses if the agent reaches its abstract goal. This reward is chosen in this proof for its generality; it simply expresses that the agent should reach $\mathcal{N}(g)$. In practice it is sparse and difficult to optimise (it is replaced in STAR by the negative distance to the center of $\mathcal{N}(g)$).

A suboptimality bound will be derived for the last $k$ steps of the trajectory, and by induction will be proven for the rest. Given criteria.4:

$$\forall s \in \mathcal{N}(g_m). |r_{\text{ext}}(g_m) - r_{\text{ext}}(s)| \leq \epsilon$$

we can claim that:

$$|V_{\pi^*_{\text{Low}}}(s_{mk}) - V_{\pi^*_{\mathcal{N}_{\text{Low}}}}(s'_{mk})| \leq \epsilon$$

To derive the upper bound on suboptimality, we will proceed by computing $V_{\pi^*_{\mathcal{N}_{\text{Low}}}}$ on the worst possible optimal trajectory $\mathcal{T}^*_{\mathcal{N}_{\text{Low}}}$. This gives:

$$|V_{\pi^*_{\text{Low}}}(s_{mk-1}) - V_{\pi^*_{\mathcal{N}_{\text{Low}}}}(s'_{mk-1})| = |r_{ext}(s_{mk-1}) + \gamma r_{ext}(s_{mk}) - r_{ext}(s'_{mk-1}) - \gamma r_{ext}(s'_{mk})|$$
$$\leq |r_{ext}(s_{mk-1}) - r_{ext}(s'_{mk-1})| + \gamma|r_{ext}(s_{mk}) - r_{ext}(s'_{mk})|$$

By triangle inequality.

The second term in the inequality corresponds exactly to $\gamma|V_{\pi^*_{\text{Low}}}(s_{mk}) - V_{\pi^*_{\mathcal{N}_{\text{Low}}}}(s'_{mk})|$ which is bounded by $\gamma\epsilon$. The first term can be expanded as such:

$$|r_{ext}(s_{mk-1}) - r_{ext}(s'_{mk-1})| = r_{ext}(s_{mk-1}) - r_{ext}(s'_{mk-1}) (\text{ since } r_{ext}(s_{mk-1}) \geq r_{ext}(s'_{mk-1}))$$
$$= -\|g^* - s_{mk-1}\|_2 + \|g^* - s'_{mk-1}\|_2$$
$$\leq -\|g^* - s_{mk-1}\|_2 + \|g^* - s_{mk-1}\|_2 + \|s_{mk-1} - s'_{mk-1}\|_2$$
$$\leq \|s_{mk-1} - s'_{mk-1}\|_2$$

---

[3] $k$ is chosen as a divisor of $M$ to simplify the notations.

Since the abstraction guarantees that starting from $\mathcal{N}(g_{m-1})$, a state $s' \in \mathcal{N}(g_m)$ is reached with $\pi^*_{\mathcal{N}_{\text{Low}}}$ in $k$ steps, then the state $s'_{mk}$ should at the worst, be reachable in one step from $s'_{mk-1}$. In addition, the state $s'_{mk-1}$ should at the worst be reachable from $s'_{(m-1)k}$ in $k-1$ steps.

In other words, the worst state $s'_{mk-1}$ is the farthest state from $g_m$ that reaches $s'_{mk}$ in 1 step and is reached by $s'_{(m-1)k}$ in $k-1$ steps. We will bound $\|s_{mk-1} - s'_{mk-1}\|_2$ using the two reachability relations.

On one hand, since $s_{mk-1}$ also reaches $g_m = g^*$ in 1 step then $\|g_{mk} - s_{mk-1}\|_2 \leq R_{max}$. Then:

$$\|s_{mk-1} - s'_{mk-1}\|_2 \leq \|s_{mk-1} - g_m\|_2 + \|g_m - s'_{mk}\|_2 + \|s'_{mk} - s'_{mk-1}\|_2$$
$$\leq 2R_{max} + \epsilon$$

On the other hand, $g_{m-1} = s_{(m-1)k}$ reaches $s_{mk-1}$ in $k-1$ steps, and then $\|g_{m-1} - s_{mk-1}\|_2 \leq (k-1)R_{max}$. Similarly, we could have:

$$\|s_{mk-1} - s'_{mk-1}\|_2 \leq \|s_{mk-1} - g_{m-1}\|_2 + \|g_{m-1} - s'_{(m-1)k}\|_2 + \|s'_{(m-1)k} - s'_{mk-1}\|_2$$
$$\leq 2(k-1)R_{max} + \|g_{m-1} - s'_{(m-1)k}\|_2$$

This results in two distinct bounds for $\|s_{mk-1} - s'_{mk-1}\|_2$ origination from a forward reachability relation, and a backwards reachability relation. This provides the following bound:

$$|V_{\pi^*_{\text{Low}}}(s_{mk-1}) - V_{\pi^*_{\mathcal{N}_{\text{Low}}}}(s'_{mk-1})| \leq \min(2R_{max} + \epsilon, 2(k-1)R_{max} + \|g_{m-1} - s'_{(m-1)k}\|_2) + \gamma\epsilon$$

Bounding $\|g_{m-1} - s'_{(m-1)k}\|_2$ depends on the nature of the abstraction, and more specifically if the abstract sets can be bounded or not. Starting from the more general case, Def.3 guarantees that $s'_{(m-1)k}$ is reachable from $s'_0$ in $m.(k-1)$ steps. To simplify the computation, can also assume that $\|s_0 - s'_0\|_2 \leq \epsilon$. (In practice, usually $s_0 = s'_0$). Also, $s_{(m-1)k}$ is reachable from $s_0$ in $m.(k-1)$ steps. Thus :

$$\|g_{m-1} - s'_{(m-1)k}\|_2 = \|g_{m-1} - s_0 + s_0 - s'_0 + s'_0 - s'_{(m-1)k}\|_2$$
$$\leq 2k(m-1)R_{max} + \epsilon$$

Then:

$$|V_{\pi^*_{\text{Low}}}(s_{mk-1}) - V_{\pi^*_{\mathcal{N}_{\text{Low}}}}(s'_{mk-1})| \leq \min(2R_{max} + \epsilon, 2(mk-1)R_{max} + \epsilon) + \gamma\epsilon$$
$$\leq 2\min(1, (mk-1))R_{max} + (\gamma+1)\epsilon$$
$$\leq 2R_{max} + (\gamma+1)\epsilon$$

With an iteration process the above reasoning can be extended: $\forall i \in \{0, \ldots, mk\}$:

$$|r_{ext}(s_i) - r_{ext}(s'_i)| = r_{ext}(s_i) - r_{ext}(s'_i)(\text{ since } r_{ext}(s_i) \geq r_{ext}(s'_i))$$
$$= -\|g^* - s_i\|_2 + \|g^* - s'_i\|_2$$
$$\leq -\|g^* - s_i\|_2 + \|g^* - s_i\|_2 + \|s_i - s'_i\|_2$$
$$\leq \|s_i - s'_i\|_2$$
$$\leq \min(2(mk-i)R_{max} + \epsilon, 2iR_{max} + \epsilon)$$
$$\leq 2\min(mk-i, i)R_{max} + \epsilon$$

with $\min(mk-i, i) = i$ if $i \leq \frac{mk}{2}$ else $\min(mk-i, i) = mk-i$.

Finally the bound can be computed as:

$$
\begin{aligned}
|V_{\pi^*_{\text{Low}}}(s_0) - V_{\pi^*_{\mathcal{N}_{\text{Low}}}}(s'_0)| &= \sum_{i=0}^{mk} \gamma^i (r_{ext}(s_i) - r_{ext}(s'_i)) \\
&\leq \sum_{i=0}^{mk} \gamma^i \|s_i - s'_i\|_2 \\
&\leq \sum_{i=0}^{mk} \gamma^i (2\min(mk-i, i)R_{max} + \epsilon) \\
&\leq \sum_{i=0}^{\frac{mk}{2}} \gamma^i (2iR_{max} + \epsilon) + \sum_{i=\frac{mk}{2}}^{mk} \gamma^i (2(mk-i)R_{max} + \epsilon) \\
&\leq \sum_{i=0}^{\frac{mk}{2}} \gamma^i (2iR_{max}) + \sum_{i=\frac{mk}{2}}^{mk} \gamma^i (2(mk-i)R_{max}) + \frac{1-\gamma^{mk+1}}{1-\gamma}\epsilon \\
&\leq \left(\sum_{i=0}^{\frac{mk}{2}} \gamma^i i + \sum_{i=\frac{mk}{2}}^{mk} \gamma^i (mk-i)\right) 2R_{max} + \frac{1-\gamma^{mk+1}}{1-\gamma}\epsilon
\end{aligned}
$$

We will now examine the case if $\exists B \geq \epsilon > 0$ such that $\forall i \in \{1, \ldots, m\}, \max_{x,y\in\mathcal{N}(g_i)}\|x-y\| \leq B$. Following the reasoning established before, we can claim that:

$$
\begin{aligned}
|r_{ext}(s_i) - r_{ext}(s'_i)| &\leq \|s_i - s'_i\|_2 \\
&\leq \min(2iR_{max} + B, 2(k-i)R_{max} + B) \\
&\leq 2(\frac{k}{2}R_{max}) + B \\
&\leq kR_{max} + B
\end{aligned}
$$

Ultimately,

$$
\begin{aligned}
|V_{\pi^*_{\text{Low}}}(s_0) - V_{\pi^*_{\mathcal{N}_{\text{Low}}}}(s'_0)| &= \sum_{i=0}^{mk} \gamma^i (r_{ext}(s_i) - r_{ext}(s'_i)) \\
&\leq \frac{1-\gamma^{mk+1}}{1-\gamma}(kR_{max} + B)
\end{aligned}
$$

### A.2 PROOF OF LEMMA 1 AND THEOREM 2:

By definition of $\mathcal{N}'$, since $G_i \in \mathcal{T}^*_{\mathcal{N}_{\text{High}}}$ then $\exists g_i \in G_i$ (since the reward for the high-level agent is $r_{\mathcal{N}_{\text{High}}}(G_i) = \max_{s\in G_i} r_{ext}(s)$) such that $g_i$ reaches a state in $\mathcal{N}_{i+1}$: To prove this we can first consider the coarsest abstraction $\mathcal{N} : \mathcal{S} \to \mathcal{G}_{\mathcal{N}} \subseteq 2^{\mathcal{S}}$ s.t $\mathcal{G}_{\mathcal{N}} = \{G_0, G_1\}$ with $G_1 = \mathcal{N}(g_m)$ (assumed to satisfy criteria 1 and 4) and $G_0 = \mathcal{N}(g_0) = \cdots = \mathcal{N}(g_{m-1})$. In this case, the goal in question $g_i$ is in $\mathcal{N}(g_{m-1})$ and by definition reaches $g_m \in G_1$. This corresponds to $g_{m-1}$. Thus, the process of refinement splits $G_0$ such as $G_0 = G' \cup G'', G' \cap G'' = \emptyset$ and identifies $G'$ such that $g_{m-1} \in G'$ (since $G'$ satisfies the reachability property for $(G', G_{i+1})$) and $G'' = \mathcal{N}'(g_0) = \cdots = \mathcal{N}'(g_{m-2})$. Additionally, $G', G''$ and $G_1$ are disjoint by definition of the refinement.

By induction, the lemma is proven. The induction also leads to a reachability-aware abstraction proving theorem 2.

## B REPRESENTATION ANALYSIS IN ANT FALL

Similarly to the study established in 5.3, we provide the results obtained in Ant Fall (visualizing the 5 dimensional representation for Ant Maze Cam is less feasible).

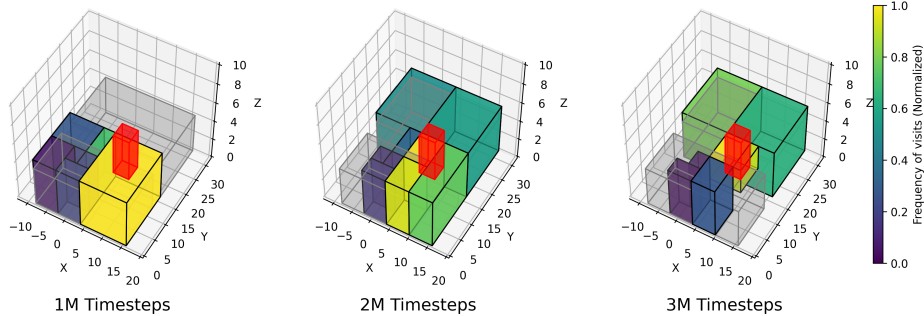

Figure 5: Frequency of goals visited by the *Commander* when evaluating a policy learned after 1M, 2M, and 3M timesteps (averaged over 5 different evaluations with 500 maximum timesteps). The subdivision of the mazes represent (abstract) goals. The color gradient represents the frequency of visits of each goal. Grey areas correspond to the platforms of the environment in Ant Fall. The red box is the movable block. Observe that, around 3M timesteps, goals are further split across the $z$-axis above the pit.

Fig.5 shows that after 1M steps the ant gradually learns to approach the area nearing the movable block (colored in yellow in the figure). Around 2M steps, the ant starts succeeding at crossing the pit by pushing the block forward. By 3M steps, the *Commander* has refined the abstraction and identified as goal the bridge-like region (immediately at the back of the movable block) that it targets most frequently to maximise its success to cross the chasm.

This representation further validates our hypothesis regarding the ability the reachability-aware abstraction to identify areas of interest in the environment and orient the learning of the agent towards optimal behaviour.

## C   ADDRESSING NON-STATIONARITY

Since STAR is a 3-level HRL algorithm, non-stationarity can affect the learning of the Tutor as well as the Commander.

To solve this problem in the Tutor, we incorporate the policy correction approach from Nachum et al. (2018) which adequately relabels subgoals in the replay buffer that are unadapted to the current policy of the agent.

Additionally, we address non-stationarity in the Commander level by only refining the abstraction when the controller's policy is stable. Precisely, reachability analysis for $(G_i, G_j)$ is only engaged when the controller's policy $\pi_{\mathcal{N}_{\text{Low}}}(s \in G_i, G_j)$ has roughly converged to a deterministic behaviour and is not randomly taking actions. Since the forward model is trained on data generated by $\pi_{\mathcal{N}_{\text{Low}}}$, it could be used as a proxy for assessing behaviour stability. Every time the transition $(G_i, G_j)$ is explored, the error $error(\mathcal{F}_k(s \in G_i, G_j))$ is evaluated and stored. Finally, we infer the stability of learned policies from the progress of the error in the forward model during a window of time; if the last 10 evaluated errors satisfy $error(\mathcal{F}_k(s \in G_i, G_j)) < \sigma$ then the policy is considered stable and reachability analysis is conducted.

## D   COMMANDER POLICY LEARNING

### D.1   POLICY TRAINING

The Commander's policy mainly uses Q-learning (Watkins & Dayan, 1992) to learn a policy $\pi_{\text{Comm}}$ for goal sampling. This choice is adapted to the discrete and small nature of the problem that the Commander is expected to solve. More precisely, both the state and action spaces of the Commander correspond to $\mathcal{G}$ since the Commander is in a state $G_t$ and has to select a next partition $G_{t+k}$ to target. Additionally, we use the learned reachability relations to orient the exploration of the agent towards reachable goals. This is done by restricting the exploration to sample reachable goals when

possible. Formally, at a goal $G_i$, the Q-values are computed only goals $G_j$ that $G_i$ is known to reach (following reachability analysis).

## D.2 HANDLING A GROWING STATE SPACE

Following a refinement of the goal space $\mathcal{G}$, a new goal space $\mathcal{G}'$ is formed such that $|\mathcal{G}'| > |\mathcal{G}|$. To handle a growing state space, the Commander transfer the Q-table computed on the state space $\mathcal{G}$ to the new state space $\mathcal{G}'$. Assuming that $G_i$ is the goal that was refined in $\mathcal{G}$, then $G'_{i_1}$ and $G'_{i_2}$ are the newly created goals with $G'_{i_1}$ reaching $G'_{i+1} = G_{i+1}$. In that case, $Q(G'_{i_1}, G'_{i+1}) = \max_{G \in \mathcal{G}'} Q(G', G'_{i+1})$ and $Q(G'_{i_2}, G'_{i+1}) = \min_{G \in \mathcal{G}'} Q(G', G'_{i+1})$. Additionally, $\forall G' \in \mathcal{G}$, s.t $G' \neq G'_{i_1}$ and $G' \neq G'_{i_2}, Q(G', G'_{i_1}) = Q(G', G'_{i_2}) = Q(G', G_i)$ (here $G = G'$). Inversely, $\forall G' \in \mathcal{G}$, s.t $G' \neq G'_{i+1} = G_{i+1}, Q(G'_{i_1}, G') = Q(G'_{i_2}, G') = Q(G_i, G')$ (here $G = G'$). In summary, the new Q-tables modifies the Q-values between the refined goal and its target while preserving the rest of the values.

## D.3 INITIALISATION OF THE ABSTRACTION

The algorithm start without an abstract goal. After initial exploration, and once the controller's policy is stable (see C for more details), we split $\mathcal{S}$ into a set of visited states and a set of unvisited ones. Consequently the initial abstraction becomes $\mathcal{G} = G_0 \cup G_0^c$, with $G_0$ the set of visited states and $G_0^c$ the complement of $G_0$ in $\mathcal{S}$, representing unvisited states. This resulting abstraction is still coarse (e.g., partitions the state once space over the $x$ and $y$ dimensions) and is mainly useful to identify an abstract state enclosing a neighborhood of the initial state.

# E ENVIRONMENT DETAILS

The implementation adapts the same environments (Ant Maze, Ant Fall) used in Nachum et al. (2018) with the addition of Ant Maze Cam. All of the environments use the Mujoco physics simulator (Todorov et al., 2012). In all the following setups, the training episode ends at $max\_timesteps = 500$. The reward signal is dense and corresponds to the negative Euclidean distance to the goal $g^*$ scaled by a factor of $0.1$. Success is achieved when this distance is less than a fixed threshold of 5.

1. **Ant Maze**: The maze is composed of immovable blocks everywhere except in $(0,0), (8,0), (16,0), (16,8), (16,16), (8,16), (0,16)$. The agent is initializes at position $(0,0)$. At each episode, a target position is sampled uniformly from $g_x^* \sim [-4,20], g_y^* \sim [-4,20]$. At evaluation time, the agent is only evaluated for a fixed exit at $(0,16)$.

2. **Ant Fall**: the agent is initialized on an elevated platform of height 4. Immovable blocks are placed everywhere except at $(-8,0), (0,0), (-8,8), (0,8), (-8,16), (0,16), (-8,24), (0,24)$. A pit is within range $[-4,12] \times [12,20]$. A movable block is placed at $(8,8)$. The agent is initialized at position $(0,0,4.5)$. At each episode, the target position is fixed to $(g_x^*, g_y^*, g_z^*) = (0,27,4.5)$. The ant has to push the block into the pit and use it as a bridge to cross to the second platform. At evaluation time, the agent is only evaluated for a fixed exit at $(0,27,4.5)$.

3. **Ant Maze Cam**: This maze is similar to Ant Maze with the addition of a new block at $(16,8)$ effectively closing any passage to the top half of the maze. For the block to be removed, the ant needs to navigate to the area in the range $[16,20] \times [0,8]$ where a camera is placed. In the area, the orientation $ori(\theta_x, \theta_y, \theta_z)$ needs to be negative simulating an identification process by the camera. $ori(\theta_x, \theta_y, \theta_z)$ is a function that projects the orientation on the $xy$ plane.

# F THE REACHABILITY ANALYSIS

The reachability analysis in STAR closely follows the process detailed in Zadem et al. (2023). First, we approximate the k-step reachability relations between states with a neural network model: $\mathcal{F}_k$. More precisely, $\mathcal{F}_k(s_t, G_j)$ predicts the state $s'_{t+k}$ reached after k steps when starting from $s_t$ and targeting the abstract goal $G_j$.

To generalise these approximations from state-wise relations to set-wise relations, we rely on off-the-shelf neural network reachability analysis techniques. Specifically, we use Ai2 (Gehr et al. (2018)) to compute the output of a neural network given a set of inputs. Consider the reachabilty analysis of a transition $(G_i, G_j)$, Ai2 computes an over-approximation of the output set of the forward model; $\tilde{R}^k_{\pi_{\mathcal{N}_{\text{Low}}}}(G_i, G_j) = \{s'_{t+k} = \mathcal{F}_k(s_t, G_j), s_t \in G_i\}$. This over-approximation is efficiently computed layer-by-layer in the neural network using operations on abstract domains. In practice, our abstract goals are represented as disjoint hyperrectangles since they are expressive in our applications and simple to analyse.

The algorithm checks if $\tilde{R}^k_{\pi_{\mathcal{N}_{\text{Low}}}}(G_i, G_j) \subseteq G_j$, i.e. the reached set of states is inside the abstract goal. If the inclusion is valid, then the reachability property is verified in $(G_i, G_j)$ and no splitting is required. Similarly, if $\tilde{R}^k_{\pi_{\mathcal{N}_{\text{Low}}}}(G_i, G_j) \cap G_j = \emptyset$ then the reachability property cannot be respected for any subset in $G_i$ and no refinement occurs. Otherwise, the algorithm splits the starting set $G_i$ in two subsets across a dimension of the hyperrectangle. Each split is recursively tested through reachability analysis and searched accordingly until a subset that respects reachability is found or a maximal splitting depth is reached.

**Abstraction Refinement:** As detailed in section F, an important step of our algorithm when conducting reachability analysis on transition $(G_i, G_j)$ is to verify if $\tilde{R}^k_{\pi_{\mathcal{N}_{\text{Low}}}}(G_i, G_j) \subseteq G_j$. In practice, since $\tilde{R}^k$ is an over-approximation, we can expect to have estimation errors in such a way that verifying the precise inclusion is very difficult. We rely on a heuristic that checks if $\frac{V(\tilde{R}^k_{\pi_{\mathcal{N}_{\text{Low}}}}(G_i, G_j) \cap G_j)}{V(G_j)} \geq \tau_1$, with $V()$ the volume of a hyperrectangle and $\tau_1$ a predefined threshold. This heuristic roughly translates to checking if most of the approximated reached states are indeed in the abstract goal. Similarly, to check if $\tilde{R}^k_{\pi_{\mathcal{N}_{\text{Low}}}}(G_i, G_j) \cap G_j = \emptyset$, we verify $\frac{V(\tilde{R}^k_{\pi_{\mathcal{N}_{\text{Low}}}}(G_i, G_j) \cap G_j)}{V(G_j)} \leq \tau_2$ with $\tau_2$ a preset parameter. That is to say, we check that most of the abstract goal is not reached. Finally, we set a minimum volume ratio of a split compared to $G_i$.

# G  HYPERPARAMETERS

## G.1  TUTOR-CONTROLLER NETWORKS

The implementation of our algorithm is adapted from the work of Zhang et al. (2023). The Tutor and Controller use the same architecture and hyperparameters as HRAC (albeit with a different goal space). It should also be noted that the policy correction method introduced in HIRO, is similarly used in HRAC and STAR. Both the Tutor and Controller use TD3 (Fujimoto et al., 2018) for learning policies with the same architecture and hyperparameters as in Zhang et al. (2023). The hyperparameters for the Tutor and Controller networks are in table 1.

## G.2  THE COMMANDER TRAINING

The Commander uses an $\epsilon$-greedy exploration policy with $\epsilon_0 = 0.99$ and $\epsilon_{min} = 0.01$ with a linear decay of a factor of $0.000001$. The learning rate of the Commander is $0.01$ for Ant Maze and Ant Maze Cam and $0.005$ for Ant Fall. The Commander's experience is stored in a buffer of size $100000$ (this buffer is used to train $\mathcal{F}_k$). The Commander's actions frequency is $k = 30$.

## G.3  FORWARD MODEL

For the forward model we use a fully connected neural network with MSE loss. The network is of size $(32, 32)$. We use the ADAM optimiser. This neural network is updated every episode for transitions $(G_i, G_j) \in \mathcal{E}$ if this transition has been sampled for a defined minimal number of Commander steps (to acquire sufficient data). Table.2 shows the hyperparameters of $\mathcal{F}_k$.

## G.4  REACHABILITY ANLYSIS

Table.3 shows the hyperparametes of the reachability analysis.

| Hyperparameters | Values | Ranges |
|---|:---:|:---:|
| Tutor (TD3) | | |
| Actor learning rate | 0.0001 | |
| Critic learning rate | 0.001 | |
| Replay buffer size | 200000 | |
| Batch size | 128 | |
| Soft update rate | 0.005 | |
| Policy update frequency | 1 | |
| $\gamma$ | 0.99 | |
| Tutor action frequency $l$ | 10 | |
| Reward scaling | 0.1 for Ant Maze (and Cam) / 1.0 for Ant Fall | $\{0.1, 1.0\}$ |
| Exploration strategy | Gaussian ($\sigma = 1.0$) | $\{1.0, 2.0\}$ |
| Controller (TD3) | | |
| Actor learning rate | 0.0001 | |
| Critic learning rate | 0.001 | |
| Replay buffer size | 200000 | |
| Batch size | 128 | |
| Soft update rate | 0.005 | |
| Policy update frequency | 1 | |
| $\gamma$ | 0.95 | |
| Reward scaling | 1.0 | |
| Exploration strategy | Gaussian ($\sigma = 1.0$) | |

Table 1: Hyperparameters for Tutor and Controller networks based on Zhang et al. (2023)

| Hyperparameters | Values |
|---|:---:|
| Forward Model | |
| learning rate | 0.001 |
| Buffer size | 100000 |
| Batch size | 64 |
| Commander steps for data-acquisition | 5000 for Ant Maze and Cam/10000 for Ant Fall |
| Epochs | 5 |

Table 2: Hyperparameters for the forward model

# H   STAR'S PSEUDO-CODE

For reproducibility, the code for STAR along with the experimental data are published in Zadem et al. (2024).

| Hyperparameters | Values |
|---|:---:|
| Reachabililty Analysis | |
| NN reachability analysis tool | Ai2 (Gehr et al., 2018) |
| $\tau_1$ | 0.7 |
| $\tau_2$ | 0.01 |
| Minimal volume ratio of a split | 0.125 |

Table 3: Hyperparameters for reachability analysis

---

**Algorithm 1** STAR

---

**Input:** Learning environment $E$.
**Output:** Computes $\pi_{\text{Comm}}$, $\pi_{\text{Tut}}$ and $\pi_{\text{Cont}}$

1: $\mathcal{D}_{\text{Commander}} \leftarrow \emptyset, \mathcal{D}_{\text{Tutor}} \leftarrow \emptyset, \mathcal{D}_{Controller} \leftarrow \emptyset, \mathcal{G} \leftarrow \mathcal{S}$
2: **for** $t \leq max\_timesteps$ **do**
3:     $\mathcal{E} \leftarrow \emptyset$
4:     $s_{\text{init}} \leftarrow$ initial state from $E$, $s_t \leftarrow s_{\text{init}}$
5:     $G_s \leftarrow G \in \mathcal{G}$ such that $s_t \in G$
6:     $G_d \sim \pi_{\text{Comm}}(G_s, g^*)$
7:     $g_t \sim \pi_{\text{Tut}}(s_t, G_d)$
8:     **while** true **do**
9:         $\mathcal{E} \leftarrow \mathcal{E} \cup \{(G_s, G_d)\}$
10:        $a_t \sim \pi_{\text{Cont}}(s_t, g_t)$
11:       $(s_{t+1}, r_t^{\text{ext}}, \text{done}) \leftarrow$ execute the action $a_t$ at $s_t$ in $E$
12:       $r_{Controller} = -\|g_t - s_t\|_2$
13:       Update $\pi_{\text{Cont}}$
14:       **if** not done **then**
15:         $s_t \leftarrow s_{t+1}, t \leftarrow t + 1$
16:         **if** $t \bmod l = 0$ **then**
17:           $r_{Commander} = -\|s_t - \text{Center}(G_d)\|_2$
18:           $\mathcal{D}_{\text{Tutor}} \leftarrow (s_{t-l}, G_d, g_{t-l}, s_t, r_{\text{Tutor}}, \text{done})$
19:           Update $\pi_{\text{Tut}}$
20:           $g_t \sim \pi_{\text{Tut}}(s_t, G_d)$
21:         **if** $t \bmod k = 0$ **then**
22:           $r_{Commander} = -\|s_t - g^*\|_2$
23:           Update $\mathcal{D}_{\text{Commander}} \leftarrow (s_{t-k}, G_d, s_t, r_{\text{Commander}}, \text{done})$
24:           Update $\pi_{\text{Comm}}$
25:           $G_s \leftarrow G \in \mathcal{G}$ such that $s_t \in G$
26:           $G_d \sim \pi_{\text{Comm}}(s_t, g_{exit})$
27:       **else**
28:         Update $\mathcal{F}_k$ with the data from $\mathcal{D}_{\text{Commander}}$
29:         $\mathcal{G} \leftarrow Refine(\mathcal{G}, \mathcal{E}, \mathcal{F}_k)$
30:         break the **while** loop and start a new episode

---

