# OpenReview forum: "Reconciling Spatial and Temporal Abstractions for Goal Representation"
_ICLR.cc/2024/Conference — ICLR 2024 poster_

### Official Review · Reviewer_awpo · 2023-10-30

**Soundness:** 2 fair
**Presentation:** 2 fair
**Contribution:** 2 fair
**Rating:** 5
**Confidence:** 4

**Summary:**

The paper introduces STAR, a hierarchical reinforcement learning algorithm. The main components of STAR are: the Navigator that selects high-level abstract subgoals, the Manager that chooses single-state subgoals that lead towards the Navigator's abstraction, and the Controller that executes low-level actions. During training, the representations are refined to better relate to the environment dynamics and agent capabilities. The authors test the performance of STAR in AntMazes.

**Strengths:**

The high-level idea of the algorithm is presented clearly, which makes the paper easier to follow. The theoretical results are coupled with their intuitive description, which despite the complex notation makes them easier to understand.

**Weaknesses:**

I doubt that the presented results are practical. My main concern is that the _Approximating the reachability property_ step seems very costly. It's a pity that the authors didn't describe this essential part in more detail, leaving only the reference to other papers (are they published anyway?). Furthermore, how do you estimate the reachability for _all states_, provided that the state space is continuous? And I think that the reachability approximation model should be analyzed as well.

The practicality issues could be addressed by adding a wall-time comparison between the presented methods. An additional discussion on the compute utilization would do as well.

I acknowledge the theoretical results, although I think that Section 4 can be restructured to be more interesting to the reader. The crucial element of this analysis is clearly Theorem 2. I suggest giving it more space and discussion. I suggest moving Theorem 1 and Lemma 1 to the appendix, as they serve only as a tool for proving Theorem 2 (as long as they are not referenced in the experiments). I would like to read here a discussion of the importance of Theorem 2, including: why does it make your method sound (recalling that you prove there exist _some refinements_, not necessarily the one you actually do in the algorithm, right?) and why it is not trivial to construct such an abstraction (e.g. by partitioning the space into any $\varepsilon$-diameter sets).

Overall, the paper lacks quite a few details, which makes it hard to understand the contribution. Since the full algorithm consists of many parts, it should be described much more clearly, not only the high-level idea. I suppose that addressing my concerns and questions should help.

I am willing to increase my rating if my concerns are addressed.

**Questions:**

How do you estimate the reachability for _all states_, provided that the state space is continuous?

What is the computational cost of refining the abstractions?

How are the abstractions represented? I bet the sets are not represented explicitly.

Why is the Navigator useful? Why not just leave the Manager and Controller? What are the theoretical and practical advantages of introducing the Navigator? Especially, given that in the end there are only several abstractions, as you show in the experiments.

Why does Theorem 2 make your method sound (recalling that you prove there exist _some refinements_, not necessarily the one you actually do in the algorithm, right?), and why it is not trivial to construct such an abstraction (e.g. by partitioning the space into any $\varepsilon$-diameter sets)?

It's a pity that by introducing the Manager you lose the abstraction of the Navigator. Can you instead of sampling a single subgoal, sample a few subgoals and somehow aggregate the low-level policy predictions to reach any of them?

Why are the proposed rewards sound? Without any justification, they look quite arbitrary.

Are there any implicit properties of the abstractions that you learn? In the experiments, they seem to be always box-shaped and disjoint, is it always the case?

Is the splitting operation the only way to refine the abstraction? Can you also incorporate the join operation if the abstractions are too fine? Is being too fine a problem in any case? Is $\mathcal N=(x\mapsto \\\{x\\\})$ always a good abstraction?

How do you handle the initial stage, when hardly any subgoal is reachable (because the agent can barely move)? Is there a danger that it will shatter the goal space arbitrarily before becoming capable of acting?

---

> ### Author Response · Authors · 2023-11-15
> **Response (part 1)**
>
> Thank you for your thorough review and questions to understand our work.
>
> Here, we answer your questions about (1) the approach’s practicality, (2) the missing algorithm’s details, and (3) the understanding of the theoretical results. We will modify our pdf to incorporate these responses to our paper.
>
> ## 1. Practicality:
>
> > How do you estimate the reachability for all states…?
>
> The reachability computation has two steps.
>
> *(A)* We approximate the k-steps reachability relation for a continuous state space with a neural network model $\mathcal{F}_k(s_t,G_d)$, called forward model (see Sec. 3.2). $\mathcal{F}_k (s_t, G_d)$ predicts the state the agent would reach in k steps starting from a state $s_t$ when applying the low level policy conditioned on the goal $G_d$ (i.e., the abstract state we want to reach). In practice, $\mathcal{F}_k$ is capable of generalizing the reachability relation to unvisited states. We train $\mathcal{F}_k$ from past explorations at the end of each episode (see Line 26 of Algorithm 1, Sec. E and Sec. D.3).
>
> *(B)* We use the set-based  reachability analysis algorithm Ai2 (T. Gehr et al., "AI2: Safety and Robustness Certification of Neural Networks with Abstract Interpretation," SP 2018 and Liu et al., 2021) on the neural network $\mathcal{F}_k$ to compute the set of states that $G_i$ can reach applying $\mathcal{F}_k$ with $G_j$ (i.e., the set $\mathcal{R} := {s’ \mid \mathcal{F}_k(s, G_j), s \in G_i}$). Such an algorithm works on continuous states via efficient **set-based operation** (i.e., all the computations
> are done on sets and not on single states). We then check if $R \cap G_j^C = \emptyset$ to check the reachability property from Definition 1.
>
> > What is the computational cost of refining the abstractions?
>
> The computational cost of the reachability analysis is negligible. We are measuring execution time on a NVIDIA RTX A5500 GPU powered computer to compare the time costs of the different environments. We are continuing to run measures of these execution times and will report in a few days the time costs over a whole run of each algorithm.
> Over 3M steps, the refinement operation (which includes the reachability analysis) takes on average 0.361s (variance 0.072, maximum 0.80s) for an oracle of 2 variables, 0.423s for 3 variables (variance 0.026, maximum 0.944s), 0.429s for 5 variables (variance 0.053, maximum 1.637).
>
> Compared to other reachability based methods (HRAC) our reachability approximation is much less costly and scalable to more variables. Running HRAC over 3M steps, we observed that the reachability relation learning step takes an increasing duration. Our measures average for the last 100 calls at 49.27s (variance:15.468 and maximum:52.206s) for 2 variables, 85.140s (variance: 12.667 max:89.940s) for 3 vars, and to 902.373s for 5 variables (the execution time continues to grow exponentially in this setting).
>
> A key difference is that the reachability relations are approximated over sets of states in STAR and not single states as in HRAC. HRAC stores all pairs of visited states in a transition matrix which is then used to repeatedly train a neural network predicting if a state reaches another. This creates a large memory constraint. The objective function of this model has to minimize the error of predicted relations compared to the transition matrix which becomes exponentially costly since it searches all the paths encoded in the matrix, which grows with more visited states (the size of the matrix increases, and the matrix is less sparse with added transition relations). On the other hand training the forward model in STAR is straightforward and uses an MSE loss to predict reached states from the replay buffer, and neural network reachability analysis extends the computation of the forward model to predict reached sets of states.
>
> > How are the abstractions represented?
>
> In our implementation each abstract state is a hyperrectangle (i.e., a multi-dimensional interval) and the abstraction is a partition of the state space that we represent with a set of *disjoint* hyperrectangles. We could substitute hyperrectangles with different abstract domains to gain precision (which we did not need), as long as the abstract states are disjoint.
>
> > Are there any implicit properties of the abstraction  …?
>
> We try to learn a reachability-aware abstraction (Definition 3).

---

> ### Author Response · Authors · 2023-11-15
> **Response (part 2)**
>
> ## 2. Algorithm details:
>
> > Why is the Navigator useful … as you show in the experiments.
>
> The 3-layer architecture with the Navigator, Manager, and Controller is key for performance and one of our main contributions.
> The Navigator learns a policy, via RL and planning, that decides what is the next *abstract state* to explore from the current state to fulfill a task. Since abstract states are finite and discrete (i.e., a *spatial abstraction*), the Navigator can solve a long-horizon task. What is difficult is to have an abstraction that “respects the lower agents dynamics”, so that the low-level policies (the Manager and Controller) manage to reach a goal $G_j$ from $G_i$. This is why we learn a reachability-aware abstraction.
> While having an abstraction with **fewer states** is desirable for easier planning for long-horizon tasks, the abstractions need to be detailed enough to represent the properties of the environment and the task, thus finding the right granularity is key. For example, in Ant Maze Fall the Navigator does not need a fine-grained subdivision of the states in the pit, but only the abstraction helping to cross the pit in correspondence with the block (see Figure 4 b). Similarly, in Ant Maze Cam having a fine-grained representation of the orientation angles most useful close to the camera position.
>
> Without the Navigator, our algorithm would not perform a spatial abstraction (similarly to HIRO).
> Figure 3 shows the practical advantages of our architecture: temporal abstraction (HIRO) and spatial abstraction (GARA) alone do not scale when increasing the dimensions *of the oracle*.
> Theoretically, Theorem 1 proves sub-optimality when using the reachability-aware abstraction.
>
> > … introducing the Manager you lose the abstraction of the Navigator. Can you instead … sample a few subgoals and … aggregate the low-level policy …?
>
> The Manager does not lose the abstraction but instead samples *a sequence of subgoals* that eventually reaches a target abstract state, acting as an intelligent tutoring system setting realistic goals for the controller (the Manager’s reward measures the success in reaching the abstract goal, see next answer).
> If we understand correctly, the question asks if the Manager can sample multiple subgoals (i.e., a breadth-first exploration). We think this strategy, differently from our implementation, will explore “closer” states first but will struggle to learn to reach a different abstract state.
>
> > Why are the proposed rewards sound? … they look quite arbitrary.
>
> We designed the rewards following the common intuition from goal-conditioned HRL algorithms: the highest-level agent in the hierarchy is rewarded when it accomplishes the task goal. Each lower-level agent is rewarded for achieving the subgoal set by its higher-level agent.
>
> The Navigator’s reward measures the negative distance from the task goal: the higher the reward, the closer is the agent to the task goal.
> The Manager’s reward measures the negative distance of the agent from the goal set by the Navigator. However, the goal in this case is a set of states (and not a single state). Thus we need to compute the distance between a point and a set. In practice, instead of computing the exact distance to a set which can be costly, we experimented with different distances (e.g., the minimum distance from a point in the set, the distance from the center of the set). We converged, after some experiments, on the distance from the center of the set.
> The Controller’s reward measures the negative distance from the subgoal state picked by the Manager, in line with the lower-level agent’s reward in HIRO, HRAC, ….
>
> > How do you handle the initial stage … before becoming capable of acting?
>
> You are correct, we cannot refine the abstraction using the forward model in the initial stages and, in general, before the low-level policies (Manager and Controller) are “stable”.
>
> In fact, we refine the abstraction *only* when the forward model approximates the reachability relation induced by the low-level policies “well enough”, and continue with the low-level policies training otherwise. In practice, we test if the forward model predicts well the reachability relation of the latest episodes (i.e., the Mean Squared Error on the prediction of the latest 10 episodes is lower than a threshold). When the test succeeds, the forward model approximates the reachability relation obtained executing the low-level policies. So, this test avoids arbitrary splittings of the goal space.
> We briefly explained this mechanism in Appendix D.4, but we will provide the above explanation in Section 3.2.

---

> ### Author Response · Authors · 2023-11-15
> **Response (part 3)**
>
> > Is the splitting operation …  good abstraction?
>
> Currently, the splitting operation is the only operation on the abstract space. We are considering join operation for future works.
> Currently, we avoid too fine abstractions through the hyperparameter “Minimal volume ratio of a split”, as mentioned in Table 3 of Annex D4. Moreover, in practice, the goal of our hierarchical representation is to drastically reduce the size of the space (from a continuous thus uncountable state space, to a finite abstraction space) to orient better exploration. Thus, with respect to this idea, a “too fine” abstraction would not be good.
>
> ## 3. Theoretical results
>
> > Why does Theorem 2 make your method sound … you prove there exist some refinements, not necessarily the one you actually do in the algorithm … and why it is not trivial to construct such an abstraction …?
>
> Thank you, we see the ambiguity in the use of “there exists” in Theorem 2. A more precise statement of Theorem 2 is:
> “Given an initial set-based abstraction $\mathcal{N}$ and assuming $\mathcal{N}(g_m)$ satisfies the Conditions (1), (2), and (3) of Definition 3, we compute a reachability-aware abstraction after a finite number of **reachability-aware refinements**.”
> Theorem 2 proves that applying the reachability-aware refinement **we use in the algorithm** for a finite number of times we compute a reachability-aware abstraction.
>
> Your observation about the existence of a trivial refinement is correct: that refinement would halve the grid in each dimension so that, for example, in 2 dimensions we’ll have 1 cell initially, then 4 cells, then 16 cells, … (a uniform grid size that is reduced by epsilon may also work). However, the trivial refinement *is not what we use in the algorithm* and it’s not useful in practice, since it does not respect the dynamic of the lower-level policies!
>
> We clarify a misunderstanding on Theorem 1, Theorem 2, and Lemma 1. Theorem 1 is important on its own and proves that, if we have a reachability-aware abstraction,  we learn a sub-optimal policy. So Theorem 1 **is not used to prove** Theorem 2.
> Then, Theorem 2 proves that the refinement computes the hypothesis of Theorem 1.
> We will improve Theorem 2 wording as above, integrate this discussion in Section 4, move Lemma 1 in the Appendix and explain Theorem 2’s proof informally.

---

> ### Comment · Reviewer_awpo · 2023-11-16
>
> Thank you for the answer.
>
> After studying Gehr et al., I understand how you propose to check the reachability. That is however hardly understandable from the paper. I suggest (1) analyzing how reliable is $\mathcal F_k$ (I suppose it's not perfect, so how accurate it is? How deep, since you have to balance its reliability and the overhead of AI2-related bounding boxes? How do you handle the moving targets issue, as the low-level policy constantly changes?), (2) switching the reference from Liu et al. to Gehr et al., (3) Providing at least a high-level idea of how the reachability is estimated. Simply stating that you use set-based operations is not enough (that's exactly what concerns me). In fact, what you do (I think) is more approximating the bounding boxes of the output sets by consecutively bounding the sets after each operation performed by the network (right?). I would like to see here even a brief (say a paragraph) summary of Sections III and IV from Gehr et al. and how you instantiate it in your algorithm (at least that you use box representations; then, focusing only on boxes and forgetting polyhedra and zonotopes in summarizing AI2 is fine). Also, I think it's important to note that in your application the true reachability sets _should_ be nearly convex, so using AI2 is even more justified than in a case of general networks.
>
> Having said that, I'm still missing how do you perform the splitting operation, how do you schedule the refinements, and how do you train the models until you get the first successful episode. Although you refer to optimal trajectories in the theorems, I believe that using sub-optimal ones can be fine in practice. However, I'm not sure what you do before the first solution is found.
>
> Now, when I understand approximating the reachability, the practicality is no longer a strong issue.
>
> Although it doesn't fit the scope of this rebuttal, I suggest comparing the performance of your approach with "Learning Multi-Level Hierarchies with Hindsight" in the future. It seems to be based on related concepts, though the actual approach differs significantly from your algorithm. It would be interesting to see a comparison.
>
> To be honest, I'm lost in the notation. Please clarify the Lemma 1. I suppose it states that after refinement the new abstraction satisfies one more reachability constraint. What does point (2) say? Why $g_j=g_i$ never works? Is everything ok with the statement of this point? Also, could you comment whether it is true that after refinement we do get one more reachabe pair, but at the same time another pair may become non-reachable (e.g. right before the splitted set)?

---

> ### Author Response · Authors · 2023-11-18
> **Response to second message (part1)**
>
> Thank you for taking the time to read our response and understand our work.
>
> > … I understand how you propose to check the reachability… That is however hardly understandable from the paper.
> > (3) Providing at least a high-level idea of how the reachability is estimated. Simply stating that you use set-based operations is not enough (that's exactly what concerns me).
> > I would like to see here even a brief (say a paragraph) summary of Sections III and IV from Gehr et al. and how you instantiate it in your algorithm (at least that you use box representations; then, focusing only on boxes and forgetting polyhedra and zonotopes in summarizing AI2 is fine).
>
> We agree, our paper misses to explain the background on reachability analysis and how we instantiate it in our context.
>
> We will add a paragraph at the end of Section 3.1 explaining how we approximate the reachable sets using abstract interpretation with Ai2, which computes an over-approximation of the network’s output layer-by-layer using operations on an abstract domain.
>
> We remark that our paper *does not provide* a novel contribution for reachability analysis of neural networks, but instead uses off-the-shelf techniques. In fact, neural network reachability analysis is a known problem with multiple algorithms and tools available (e.g., see the (Verification of Neural Network competition)[https://sites.google.com/view/vnn2023] organized during CAV and AAAI).
>
> > In fact, what you do (I think) is more approximating the bounding boxes of the output sets by consecutively bounding the sets after each operation performed by the network (right?).
>
> Yes! Your understanding about over-approximating the neural network outputs is correct.
>
> > I suggest (1) analyzing how reliable is $F_k$  (I suppose it's not perfect, so how accurate it is? How deep, since you have to balance its reliability and the overhead of AI2-related bounding boxes? )
>
> In the algorithm, we try to train a precise $\mathcal{F}_k$: we use $\mathcal{F}_k$ in the reachability analysis only when it accurately approximates the data we observed so far (see last answer from (our second response)[https://openreview.net/forum?id=odY3PkI5VB&noteId=uoi64FiwJ0]).
>
> Yes, there are tradeoffs between the size of the $\mathcal{F}_k$ model (see Appendix D.3), its precision, and the overhead of running reachability analysis. The architecture of $\mathcal{F}_k$ is a hyperparameter that we decided via hyper parametrization on the Ant Maze environment.
>
> > how do you perform the splitting operation,
>
> We will add a high-level description of the splitting at the end of Section 3.1, and a more in depth description of the procedure in the Appendix.
>
> The input to the refinement is a sequence of visited abstract goals $G_0, \ldots, G_n$.
> For each pair $G_i, G_j$ in the sequence, the refinement computes a subset $G_\text{split}$ of states in $G_i$. Each state $s \in G_\text{split}$ is such that $\mathcal{F}_k(s, G_j) \in G_j$, and
>
> each state $s \in G_{\text{split}}^C$ ($G_\text{split}$’s complement) will not reach $G_j$. Then, the refinement removes $G_i$ from the abstraction, adding $G_{\text{split}}$ and $G_{\text{split}}^C$.
>
> In practice, since we use intervals, which are convex, we find *a set of subsets* $\mathcal{G}_{\text{reach}}$ of $G_i$ that contains states that reach $G_j$.
>
> We use the same splitting mechanism from (Shiqi Wang, Kexin Pei, Justin Whitehouse, Junfeng Yang, Suman Jana. Formal Security Analysis of Neural Networks using Symbolic Intervals. USENIX Security Symposium 2018)  and GARA (Zadem et al., 2023).
> At the high level, the algorithm recursively splits a set $G_s$ (initially, it is $G_i$) to find such subsets by:
>
> 1.Computing the over-approximation $G_R$ of the states reachable from $G_s$ using Ai2.
>
> 2.Checking if $G_R \subseteq G_j$ to see if $G_s$ should be part of $\mathcal{G}_\text{reach}$.
>
> If it is the case, $G_R$ satisfies the reachability property (Definition 1) and is added to $\mathcal{G}_\text{reach}$.
>
> 3.Otherwise, checking if $G_R \subseteq G_j$ is empty. In that case, the algorithm stops.
>
> 4.Otherwise, the algorithm splits $G_s$ in two, along one of the interval dimensions (with the highest sensibility on the output, measured via its gradient) , and recursively applies the same procedure to each one of the two new sets.
>
> In practice, during step 2 and 3 we do not check exactly that $G_R \subseteq G_j$ but instead use an approximation: we compute the ratio $\frac{\text{volume}(G_R \cap G_j)}{\text{volume}(G_j)}$, and if that is smaller than a threshold (the hyperparameter $\tau1$ we show in Table 3) we add the set to $\mathcal{G}_\text{reach}$. This translates to checking if most of $G_R$ is inside $G_j$.

---

> ### Author Response · Authors · 2023-11-18
> **Response to second message (part 2)**
>
> > how do you schedule the refinements
> >  How do you handle the moving targets issue, as the low-level policy constantly changes?
>
> We answer this question in (this)[https://openreview.net/forum?id=odY3PkI5VB&noteId=rRpjSVPgLh] recent answer to reviewer ep9K, which clarifies how the refinement copes with * High-variance during training*.
>
> > how do you train the models until you get the first successful episode.
>
> We describe the “Abstract state initialization” heuristic we use in (this)[https://openreview.net/forum?id=odY3PkI5VB&noteId=rRpjSVPgLh] recent answer to reviewer ep9K.
> Note we don’t need to get a successful episode (i.e., reaching the task goal $g^*$) to refine the abstraction, but we need to train a “stable” forward model $\mathcal{F}_K$.
>
> Although you refer to optimal trajectories in the theorems, I believe that using sub-optimal ones can be fine in practice. However, I'm not sure what you do before the first solution is found.
>
> Now, when I understand approximating the reachability, the practicality is no longer a strong issue.
>
> > … I suggest comparing the performance of your approach with "Learning Multi-Level Hierarchies with Hindsight" in the future …  related concepts
>
> Yes, thank you for this suggestion. For this work, we decided to compare with HRAC since it captures similar state-wise reachability relations (albeit more efficiently) to Levy et al. 2019 to drive the learning. We will consider adding this in our future comparisons.
>
>
> > Please clarify the Lemma 1. I suppose it states that after refinement the new abstraction satisfies one more reachability constraint.
>
> Thank you for reading Lemma 1 carefully! Yes, your intuition on the Lemma is correct.
>
> When answering your questions we realized that the statements of Lemma 1 and Theorem 2 had a typo in the required pre-conditions on $\mathcal{N}$ (in one of our later paper revision we refactored Definition 3 - reachability-aware abstraction - but we did not correctly update Lemma 1 and Theorem 2).
>
> Instead of `satisfies the Conditions (1), (2), and (3) (but not (4)) of a reachability-aware abstraction (Definition 3)`, the Lemma just needs condition (1) and (4) from Definition 3 (and the other condition we state that $\mathcal{N}’$ is a reachability-aware refinement of $\mathcal{N}$). Requiring Conditions (2) and (3) is too strong and is not needed in the Lemma’s proof (which instead requires (4)).
>
> We apologize for the oversight, we updated the paper’s pdf applying the correct fixes (you can see the old text is strikeout and the diff from our previous version is colored in blue).
>
> >  What does point (2) say? Why $g_i=g_j$  never works?
>
> Point 2 in Definition 3 implies that in a reachability-aware abstraction, all optimal goals should belong to disjoint abstract sets. This property plays a critical role in proving the regret bound for the policy in Theorem 1. Intuitively, it results in the optimal high level policy $\pi_{\mathcal{N}_{High}}$ having a trajectory that visits the abstract sets containing g_0, g_1, … which correspond to neighborhoods around these points (where there is a bound on the reward).
>
> Point 2 in Lemma 1 specifies that the new abstraction separates at least two optimal subgoals $g_i, g_j \in \mathcal{T}_{\text{high}}^*$ (concrete states) into disjoint new abstract states, satisfying property 2 in Definition 3 in one more pair.
>
> > Also, could you comment whether it is true that after refinement we do get one more reachable pair, but at the same time another pair may become non-reachable (e.g. right before the splitted set)?
>
> Yes that is correct and only possible for incoming reachability relations to the split abstract goal. In such cases the algorithm simply re-examines the modified relations in the later episodes and refines if necessary.

---

> ### Comment · Reviewer_awpo · 2023-11-21
>
> Thank you for the answer.
>
> Based on your clarifications, I understand your approach much better, thus I can increase my rating. However, in my opinion it is essential to include the clarifications we discussed to the main body of the paper, which may require considerable changes compared to the current state of the paper.

---

### Official Review · Reviewer_ep9K · 2023-11-01

**Soundness:** 2 fair
**Presentation:** 2 fair
**Contribution:** 3 good
**Rating:** 6
**Confidence:** 4

**Summary:**

In this paper, the authors introduce a groundbreaking three-layer Hierarchical Reinforcement Learning (HRL) algorithm known as STAR, designed to tackle complex tasks by combining spatial and temporal abstractions in goal representation. Goal-conditioned HRL has proven effective in breaking down challenging tasks into simpler subtasks, but previous methods encountered limitations when dealing with environments characterized by intricate state reachability relations. STAR, on the other hand, addresses these challenges by introducing both temporal and spatial abstractions, presenting a novel approach that bridges the gap between these two essential aspects of HRL.

Additionally, they provide theoretical insights into the regret bounds of learned policies. Furthermore, the authors empirically demonstrate the power of STAR in complex continuous control tasks, showing its ability to scale to environments with intricate dynamics. This work is said to contribute three elements to the field of HRL: the novel STAR algorithm with its three-layer hierarchy, the theoretical justification for reachability-aware goal representations, and empirical evidence of STAR's effectiveness in combining temporal and spatial abstractions to handle complex tasks. These contributions position STAR as a solution for addressing the challenges posed by complex, high-dimensional environments in reinforcement learning.

**Strengths:**

- STAR performs online learning of both policies and representations.
- Using reachability provides meaningful goal representations by exploiting the dynamics of the environment, and allows scaling up to complex continuous state space control problems.
- Theoretical contributions: The authors attempted to define a bound on the sub-optimality of policies trained with reachability-aware abstractions, reinforcing their approach's theoretical basis, and providing support for the progressive refinement of these abstractions during the learning process. (Question 3)
- Reasonable choices of environment settings and baselines to evaluate the proposed approach.

Writing skills:
- In addition to explaining the intuition behind the proposed solution, the authors have nicely put it into a mathematical description that is also easy to follow and understand.

**Weaknesses:**

Majors:
- The results have been plotted only for 5 runs. Usually, this is not enough number of runs. Especially, in AntMaze and AntMazeCam where the results of HIRO and HRAC are close to STAR.

- It is not clear what a "Complex" continuous control task means. How to measure the complexity? Is it only related to scaling up the dimensionality, or is it also affected by distribution changes or even complications in the environment?

- It seems important to have a deterministic environment. The paper mentions that STAR is able to perform online. I assume it might be able to adjust to the stochasticity of the environment with some tricks.

- I suspect this approach might suffer from high variance during learning due to the high non-stationarity between multiple layers of hierarchical agents. However, I believe there exist some tricks to alleviate the problem. For example, define the reachability for a backward model, so that it starts from the Goal set that the task goal $g*$ belongs to, and find all groups of states that FROM them, $g*$ group is reachable. This way, we might not need to go over all G_0 to G_n subset of abstract goal sets visited in an episode, but just visit the ones that lead to the G* set.

- I think there are two types of policies based on the way you are using them in this paper. One type of policy maps state to state (maybe a policy-conditioned transition probability?), and goals can be sampled from them; the other type is the common concept of policy in RL, where maps state to actions. See Q1 and Q6, please.


Minors:
- imprecise usage of motifs: see questions 1 for example
- And some minor typos like: e.g., in Conclusion, ln 2: 'spactial' instead of 'spatial*', in 5.3: 'In Fig.4 we we',

All in all, I have doubts about the soundness of the conclusions and proofs in the paper. I would be happy to change my rating if I learn more about my questions. I see the value of the core idea of this work.

**Questions:**

1- In section 3-1, at the end of manager and navigator definitions, what do you exactly mean by $Gt+k ∼ \pi_{Nav}(s_t, g∗)$ and $g_{t+l} ∼ \pi_{Man}(st, G_{t+k})$? Is not '~' used to show sampling from a distribution everywhere else in the paper? I assume it is only meant to say the abstract goal set and subgoals are conditioned on $\pi_{Nav}$ and $\pi_{Man}$, respectively.

2- How is the composition of $\pi_{Man}$ and $\pi_{Cont}$ to generate $\pi_{low}$?

3- How do you initialize $N$ and a state set $G$ of interest to solve the task? Is there a measurement for that or is it randomly chosen based on $pi_{nav}$? If it is based on $pi+{nav}$ behavior, how do you deal with the high variance during training? Especially when the dimensionality of the environment increases, variance exponentially grows.

4- The Manager's reward can help in simple cases to learn how to sample subgoals that help the agent reach Gt+k. But if there are fixed randomly shaped barriers in an environment, is this still working? Planning might help with such cases. I think it is important to design the Manager's reward carefully based on the environment's dynamics.

5- Are all the proofs in section 4 novel, or did they exist in previous work (e.g. in GARA or Liu 2021)?

6- "....  $g ∼ \pi^∗_{high}(s, g^∗)$ that samples $g \in S$, and a low-level policy $a ∼ π^∗_low(s, g)$ that samples actions $a \in A$", how come that both actions and goals are sampled from policies?

---

> ### Author Response · Authors · 2023-11-16
> **Response (part1)**
>
> We thank you for the insightful review and all the feedback on our work. In the following, we first clarify some points from the weaknesses section, and then answer your questions.  We will modify our pdf to incorporate these clarifications.
>
> ## Clarifications
>
> > The results have been plotted only for 5 runs … not enough…
>
> We agree the confidence in our results could increase with more runs.
>
> We’re running more experiments and plan to obtain about 8 runs before the discussion ends for AntMaze (STAR, HIRO, HRAC algorithms) and AntMazeCam (STAR, HIRO algorithms) and then update the pdf accordingly (runs are computationally expensive *for all the algorithms*).
>
> We will collect up to 10 runs for all the environments and algorithms before the paper acceptance deadline (both the HIRO and LESSONS algorithms provide an evaluation with 10 runs).
>
> > … what a "Complex" continuous control task means. How to measure the complexity? … affected by distribution changes …?
>
> Indeed, task complexity is an ill-defined notion and no consensus has been reached on a single measure of task complexity, as there are many orthogonals measures to be considered. In our case, the complexity is linked to (1) scaling up the dimensionality of the state and action spaces and, in particular, the dimension of the oracle state space, as we show that we scale to 5 dimensions, compared to 2 in the SOTA; (2) the sparse reward function which translates the idea of a long-horizon task. We do not explicitly consider distribution changes.
>
> > It seems important to have a deterministic environment. … adjust to the stochasticity … with some tricks.
>
> STAR does not have a specific mechanism to cope with stochasticity. *We share* the deterministic environment assumption with HIRO and LESSONS. Only HRAC showed to be robust to small perturbations in the environment, which is a limited form of stochasticity.
>
> We hypothesize STAR’s abstraction to be also robust to small perturbations, since they would not drastically change the reachability relation among goals. However, we think STAR’s abstraction and reachability analysis should be extended (e.g., to quantify the environment uncertainty) to handle stochastic environments.
>
> ## Answers to direct questions
>
> > 1. In section 3.1, … Is not '~' used to show sampling from a distribution …?
> > I think there are two types of policies based on the way you are using them in this paper….
>
> > 6. … how come that both actions and goals are sampled from policies?
>
> We thank you for highlighting the imprecision with respect to the use of '~'; we improperly used it to show sampling from a policy, be it stochastic or deterministic.
>
> We clarify the input and output of each of the goal-conditioned policies in STAR:
>
> $\pi_{Nav}: \mathcal{S} \times \mathcal{S} \rightarrow \mathcal{G}$. This policy takes a state $s_t$, and a task goal $g^*$ and outputs an abstract goal $G_{t+k}$.
>
> $\pi_{Man}: \mathcal{S} \times \mathcal{G} \rightarrow \mathcal{S}$. This policy takes a state $s_t$, and an abstract goal $G_{t+k}$ and outputs a subgoal $g_{t+l}$.
>
> $\pi_{Cont}: \mathcal{S} \times \mathcal{S} \rightarrow \mathcal{A}$. This policy takes a state $s_t$, and a subgoal $g_{t+l}$ and outputs an action $a_t$.
>
> Similarly we’d like to clarify the goal-conditioned policies used in the theoretical sections:
>
> $\pi^*_{high}: \mathcal{S} \times \mathcal{S} \rightarrow \mathcal{S}$. This policy takes a state $s_t$, and a task goal $g^*$ and outputs a concrete goal (state) $g_{t+k}$.
>
> $\pi^*_{low}: \mathcal{S} \times \mathcal{S} \rightarrow \mathcal{A}$. This policy takes a state $s_t$, and a subgoal $g_{t+k}$ and outputs an action $a_t$.
>
> $\pi^*_\mathcal{N, High}: \mathcal{S} \times \mathcal{S} \rightarrow \mathcal{G}$. This policy takes a state $s_t$, and a task goal $g^*$ and outputs an abstract goal $G_{t+k}$.
>
> $\pi^*_\mathcal{N, Low}: \mathcal{S} \times \mathcal{G} \rightarrow \mathcal{A}$. This policy takes a state $s_t$, and an abstract goal $G_{t+k}$ and outputs an action $a_t$.
>
> Such hierarchical policies that output goals of different levels of abstractions are standard in goal-conditioned HRL with a 2-level hierarchy (e.g., see HIRO, HRAC, LESSON).
>
> > 2. How is the composition of $\pi_{\text{Man}}$ and $\pi_{\text{Cont}}$ to generate  $\pi_{\text{low}}$?
>
> Focusing on the composition of $\pi_{\text{Cont}}$ and $\pi_{\text{Man}}$ to generate a hierarchical policy $\pi_{\text{low}}$, we can write:
>
> $\pi_{\text{low}}(s_t, G_{k*[ t/k ])}) := \pi_{\text{Cont}}(s_t,\pi_{\text{Man}}(s_{l*[ t/l ]},G_{k*[ t/k ]}))$
>
> where $[n]$ computes the floor of the number $n$.

---

> > ### Author Response · Authors · 2023-11-16
> > **Response (part2)**
> >
> > > 3. How do you initialize $\mathcal{N}$ and a state set $\mathcal{G}$ …  to solve the task? … measurement … or is it randomly chosen based on $\pi_{\text{Nav}}$?
> > > … how do you deal with the high variance during training?
> >
> > *Abstract state initialization*: we have an initialization step where we (a) perform some random explorations (still without an abstract goal); (b) train a forward model (see [first answer to Reviewer awpo here](https://openreview.net/forum?id=odY3PkI5VB&noteId=i1r1FtB3Ue)) to approximate the reachability relation of the explored data; and (c) once the forward model is precise (see the second part of this answer) we obtain an initial abstraction separating the reachable and unreachable states (in the data). This resulting abstraction is still coarse (e.g., partitions the state once space over the x and y dimensions) and is mainly useful to identify an abstract state enclosing a neighborhood of the initial environment state.
> >
> > *High-variance during training*: we mitigate this problem, which may cause non-stationarity of policies at different levels, both when refining the abstraction and in the Manager.
> >
> > We refine the abstraction at the end of every episode **only if** the forward model approximates the reachability relation induced by the low-level policies. In practice, we test if the forward model predicts “well” the reachability relation of the latest episodes (testing if the Mean Squared Error on the prediction of the latest 10 episodes is lower than a threshold, see Section D.4). We hypothesize that, when the above test succeeds, the low-level policies are *stable* and the forward model captures the reachability relation. This heuristics helps in coping with the non-stationarity issue.
> >
> > We further thank you for your suggestion about the refinement strategy (i.e., the “trick” to refine in a backward direction from the goal abstract state): we agree it may help in coping with non-stationarity and it is worth exploring in the future.
> >
> > We further use off-policy correction (Nachum et al., 2018) to deal with non-stationarity in the Manager agent.
> >
> > > 4. The Manager's reward  … help the agent reach Gt+k. But if there are fixed randomly shaped barriers in an environment …? …  design the Manager's reward carefully based on the environment's dynamics.
> >
> > We apologize, we do not understand precisely the scenario with obstacles you describe in the question (e.g., are the barriers randomly appearing in the environment or are the barriers fixed and just with an “unusual” shape? What would be a concrete example of such an environment?).  We’d be glad if you could clarify this question.
> >
> > At the high level, the Manager’s reward measures the distance from a goal selected by the Navigator (see [third answer here](https://openreview.net/forum?id=odY3PkI5VB&noteId=uoi64FiwJ0)).
> >
> > > 5. Are all the proofs … novel … ?
> >
> > *Yes, all the proofs are novel*: GARA (Zadem et al., 2023) does not have proofs and the proofs from LESSONS (Li 2021) are specific to learning goal representations using “slow features”.

---

> > > ### Author Response · Authors · 2023-11-20
> > > **Update to response**
> > >
> > > We have updated our comparative analysis with more runs for the critical approaches with close performances (We added the new plot in the pdf). AntMaze is now computed over 8 runs for STAR, HIRO and HRAC. AntMazeCam is also computed over 8 runs for STAR and HIRO algorithms. We are aiming to collect 10 runs for all the baselines before the deadline of acceptance.

---

> > > > ### Comment · Reviewer_ep9K · 2023-11-22
> > > >
> > > > Thanks for your responses Part and Part 2,  and the new results of 8 runs. They were clarifying and helpful! I believe it would be nice for the idea of this paper to be out there. Therefore, I increase my rating. However, this algorithm still needs to be tried on more complicated environments. Pointing back to my question 4, for environments such as Pinball with complex obstacles, the performance might be very much dependent on the careful design of the reward function.

---

### Official Review · Reviewer_L6eb · 2023-11-01

**Soundness:** 3 good
**Presentation:** 2 fair
**Contribution:** 3 good
**Rating:** 8
**Confidence:** 3

**Summary:**

The paper addresses reachability-aware abstraction with temporal abstraction in goal representation for Hierarchical Reinforcement Learning. The proposed model, building from ideas described in GARA(Zadem et al., 2023), tries to overcome the limitations of SOTA methods suffering from scalability when dealing with complex goals or high-dimensional environments. \
STAR introduces a three-agent architecture where every agent acts on distinct timescales with separate reward functions. The three layers feed different goal abstractions, subject to a reachability condition that refines them. Under the assumption that the environment is deterministic and the reward signal is bound in the environment, the authors show that the newly refined abstraction leads to a bound on the sub-optimality of the hierarchical policy.\
The experiments are three of increasing dimensionality, namely bi-dimensional Ant Maze, tri-dimensional Ant Maze Fall, and five-dimensional Ant Maze Cam. Results show that the method proposed achieves a higher success rate in fewer time steps w.r.t. three recent HRL methods.\
 In the appendices, the authors report proofs of the theorems appearing in the main text, details of the architecture, specifications of the experiments, and STAR pseudo-code.

**Strengths:**

Originality:\
The idea is original, as far as the reviewer knows, though limited to 'an extension of HIRO with the addition of a reachability component', as the authors note. The idea of abstraction based on a recursive splitting states partition to refine the agents' dynamics is new, and it ensures reachability for each agent at different levels of abstraction.


Quality:\
As far as the reviewer could check, the paper provides sound statements that support the implementation and the results. There are a few misprints that can be easily checked.

Clarity:\
The paper has a good structure; it is quite straightforward, and it explains nicely the problem and its collocation in the literature. The proofs in the appendix are reasonably clear, and the STAR pseudo code supports a good procedural understanding of the proposed method.


Significance:\
The experiments improve the results over the chosen HRL competitors.   The paper provides a contribution to HRL with a simple yet effective idea.

**Weaknesses:**

The goal space {\mathcal G} and the state space S need to be more clearly stated. By definition, {\mathcal G} is a partition of S, initially a coarse one. In B.1. of the Appendix, it is written that "both the state and action spaces of the Navigator correspond to {\mathcal G}".

 What does not seem clear is:\
1. What are initially {\mathcal G} and G_t.  Namely, if {\mathcal G} initially contain the whole set S and not just some subset of S, then it is clear that the next partition is chosen from {\mathcal G} or from the current G_t, otherwise, some clarification on the initial {\mathcal G} might be needed, also for actions.
2. The action space is said to be in {\mathcal G}, and it needs to be explained if it is in {\mathcal G} regarding the history or possible actions. Thus, it would be better to clarify R_{max}.

Maybe I missed something, but I do not see how just assuming
{\mathcal N}(g_m), with the required conditions of Definition 3, cannot happen at a splitting point of a chosen  G_i that no state satisfies the reachability property. In particular, what is M in the last but one line on page 6?

The sentence on page 4, after the Navigator reward, "The max in this reward is computed over observed exploration data", is not entirely clear, "observed" in which sense?

In Definition 2,  after removing G_i from {\mathcal G}_{\mathcal  N} and adding G_1' and G_2', why is it necessary to specify that G_i is the union of the two after G_i has been removed and since only G_1' is chosen.

It seems that to do the exploration for checking the reachability property (page 5), the complete partition of the states is required, but how the exploration scales with the dimension of S is not discussed.
Appendix B.2 discusses the growing state space, but there is no discussion about the computational cost of the exploration given the state space partition.

Figure 4 is not so exciting; I do not see that "progressively, the ant explores trajectories leading to the goal of the task."
I found the representation of the exploration of the Ant Maze environment given in Nachum et Al. ICLR 2019 quite appealing. The pictures could be improved.

**Questions:**

1. Have STAR been experimented on different tasks than the maze?
2. What network is used to train {\mathcal F}_K?
3. Why not compare with Nachum et Al. ICLR 2019?
4. Have STAR been experimented with larger scale space than the example given?
5. Can you estimate the trade-off between results and resources (e.g. memory usage, algorithm time cost, etc.) for STAR compared to the other methods?

---

> ### Author Response · Authors · 2023-11-17
> **Response (part 1)**
>
> We thank you for your careful review and insightful feedback, we will use that to improve the clarity of our paper.
>
> ## Clarifications
>
> > What are initially $\mathcal{G}$ and $G_t$ …
> > … action space … explained if it is in {\mathcal G} regarding the history or possible actions.
>
> $\mathcal{G}$ is *a partition* of the environment’s space $S$:  that is, $\mathcal{G}$ is a set of states $\{G_1, \ldots, G_n\}$, where each $G \in \mathcal{G}$, $G \subseteq S$, all the subsets are disjoint (for any $i \neq j$,  $G_i \cap G_j = \emptyset$ and the union of the subsets cover $S$, i.e., $\bigcap_{G \in \mathcal{G}} = S$).
>
> The Navigator policy $\pi_{Nav}$ takes as input the current environment state $s_t \in S$ and a task goal state $g^* \in S$ (this is the final environment’s state the agent should reach), and outputs a goal state $G \in \mathcal{G}$. So, the action space of the Navigator is also $\mathcal{G}$.
>
> In practice,  we initialize the partition $\mathcal{G}$ as explained in the last answer in this [message](https://openreview.net/forum?id=odY3PkI5VB&noteId=uoi64FiwJ0) to reviewer ep9K.
>
> > clarify $R_{max}$
>
> $R_{\text{max}}$ is the maximum distance between two states in the environment that are reachable in 1 step, and this value bounds the reward signals at each step.
>
> > Maybe I missed something, but I do not see how just assuming $\mathcal {N}(g_m)$, with the required conditions of Definition 3, cannot happen at a splitting point of a chosen $G_i$ that no state satisfies the reachability property.
>
> We do not understand this question completely, we’d be glad  if you could provide more context.
>
> Here, we try to answer explaining what happens during a refinement. We have a sequence of abstract states $G_1, G_2, \ldots $ that we visited in the last episode.  We check the reachability property between every $G_i, G_{i+1}$ in the sequence. If the reachability property does not hold, then we split the abstract state, otherwise not. If all the pair of states satisfy the reachability property, then we do not split the abstraction.
>
> Then, in Lemma 1, we show that if $\mathcal{N’}$ is a reachability-aware refinement of  $\mathcal{N}$, then it contains a new abstract state $G_i$ that “separates” points from the optimal high-level trajectory.
>
> > In particular, what is $M$ in the last but one line on page 6?
>
> $M$ is the deterministic environment.
>
> > "The max in this reward is computed over observed exploration data", is not entirely clear, "observed" in which sense?
>
> The Navigator’s reward for $s_t \in G_{t+k}$ is the maximal reward obtained by any state visited inside $G_{t+k}$ during the past $k$ steps. Since the Navigator operates every $k$ steps, multiple (environment) states can be visited inside $G_{t+k}$ before the reward of the Navigator $r_{Navigator}$ is computed.
>
> > In Definition 2, after removing $G_i$ from $\mathcal{G_{\mathcal N}}$ and adding $G_1'$ and $G_2'$, why is it necessary to specify that $G_i$ is the union of the two after $G_i$ has been removed and since only $G_1'$ is chosen.
>
> This operation is motivated by our initial presentation of the abstraction as a **partition** of the state space $\mathcal{S}$.
>
> It is true that $G’_1$ is the one that reaches $G_j$, but $G’_2$ is not discarded as a potential abstract goal that can be explored by the agents (i.e., it is still part of the partition).
>
> > … how the exploration scales with the dimension of $S$ is not discussed … there is no discussion about the computational cost of the exploration given the state space partition.
>
> The exploration using the abstract state space is challenging in high-dimensional environments (i.e., it is difficult to explore an abstract state from another one). This is in fact the main issue of the GARA algorithm that we address in STAR by adding a temporal abstraction mechanism.
>
> > Figure 4 … I do not see that "progressively, the ant explores trajectories leading to the goal of the task … representation in Nachum et al. ICLR 2019 quite appealing.
>
> We see the representation from Figure 2 in (Nachum et al., ICLR 2019): we will provide a similar figure for the policy we learn after 3M timesteps. Note that Figure 2 in  (Nachum et al., ICLR 2019) does not show how the abstract goal representation evolves during training.

---

> > ### Comment · Reviewer_L6eb · 2023-11-21
> > **Thank you**
> >
> > Thank you for your answer.

---

> ### Author Response · Authors · 2023-11-17
> **Response (part 2)**
>
> ## Answers to direct questions
>
> > 1. Have STAR been experimented on different tasks than the maze?
> >  4. Have STAR been experimented with larger scale space than the example given?
>
> No, STAR has been experimented on the different variants of the maze environment described in the experiments.
>
> > 2. What network is used to train $\mathcal{F}_K$?
>
> $\mathcal{F}_K$ is a connected neural network with (32 x 32) size (see appendix D.3, and Table 2 for the network’s hyperparameters).
>
> > 3. Why not compare with Nachum et Al. ICLR 2019?
>
> We agree, the comparison could be interesting. We think HIRO, HRAC, and LESSONS to be the most significant baselines for evaluating our research questions: they implement temporal abstraction with different goal representations (a subset of the environment’s state space, an embedding representing a reachability relation, and a latent space via slow features respectively).
>
> > 5. Can you estimate the trade-off between results and resources (e.g. memory usage, algorithm time cost, etc.) for STAR compared to the other methods?
>
> We are collecting the data for estimating the run times of the different algorithms and will report them in the pdf as soon as we have them.

---

> > ### Author Response · Authors · 2023-11-20
> > **Update to response**
> >
> > We have conducted a preliminary study of resources used by the critical approaches with close performances in AntMaze and AntMazeCam, and we report runtimes (measured in seconds) and reachability learning time cost (if applicable), averaged over 3 runs.
> >
> > for AntMaze:
> > - STAR: 32287s, variance: 412, the reachability analysis takes 0.01% of the total runtime
> > - HRAC: 38997s, variance: 302, the reachability analysis takes 15.33% of the total runtime
> > - HIRO: 30189s, variance: 343
> >
> > for AntMazeCam:
> > - STAR: 32760s, variance: 312, the reachability analysis takes 0.013% of the total runtime
> > - HRAC: we did not measure this, this approach however is very costly on this setting since it does not scale well to higher dimensions. Learning reachability relations between states requires an exponentially large search.
> > - HIRO: 30999s, variance: 274

---

### Author Response · Authors · 2023-11-23
**Message for everyone**

Dear reviewers, we earnestly  thank you for the valuable feedback and discussions. We are convinced it helped us improve the quality of our paper.
We submitted a new paper’s version to address the most pressing feedback (and we will address the others after the discussion period):

**Not enough runs to trust the results:** we increased the number of runs and have included in the pdf the results of 8 runs (instead of 5) for the approaches that had close performance (STAR, HRAC and HIRO in Ant Maze, STAR and HIRO in Ant Maze Cam).  We stress that the paper results *did not change adding more runs*, validating the conclusions of our experiments. We continue to collect more runs for all the environments and approaches that we will include the results for 10 runs in the final paper’s version.


**Missing details about reachability computation and refinement:** we added subsection 3.2.1 explaining: (a) how we approximate reachability and when we refine the abstraction to avoid non-stationarity issues in the Navigator; (b) how we use abstract interpretation to approximate a set of reachable states for checking pairwise reachability; and (c) how we split the abstraction.
We think this additional section provides the missing background in abstract interpretation and clarifies how we compute a refinement.
To gain the necessary space for this subsection, we moved the representation analysis for Ant Fall (former Figure 4b) in the Appendix B. Figure 4 still shows the qualitative analysis for Ant Maze (which is similar to Ant Fall).

**Theorem 2 clarifications:** we rephrased Theorem 2’s statement and fixed the typos in both Lemma 1 and Theorem 2.

**Changes to the appendix:** we moved the representation analysis of ant fall in appendix B which will now serve as an extension of the study presented in 5.3. In appendix C we clarify how non-stationarity is handled in our algorithm. Appendix D.3 describes how the abstraction is initialized. Appendix F provides technical details regarding reachability analysis and explains the used heuristics.

---

### Meta-Review · Area_Chair_nGmB · 2024-01-10

**Metareview:**

This paper introduces a novel hierarchical reinforcement learning architecture. Fundamentally, the proposed approach is a 3-levels Feudal HRL algorithm capable of performing both spatial and temporal abstraction.

This is a timely and interesting contribution to the RL literature.

All the reviewers agree that the algorithm is interesting, and worth of publication.

The authors should make sure to incorporate the remaining feedback from the reviewers that has not yet been considered in the updated manuscript.

**Justification For Why Not Higher Score:**

The thoroughness of the experimental evaluation, and the clarity of the manuscript could be further improved, consistently with the feedback of the reviewers.

**Justification For Why Not Lower Score:**

All the reviewers agree that the paper is interesting and worth publication.

---

### Decision · Program_Chairs · 2024-01-16

Accept (poster)